# Surface-mediated bacteriophage defense incurs fitness tradeoffs for interbacterial antagonism

Chia-En Tsai[1,2,5], Feng-Qi Wang [ID] [2,5], Chih-Wen Yang[3], Ling-Li Yang[2], Thao VP Nguyen [ID] [2], Yung-Chih Chen [ID] [2], Po-Yin Chen [ID] [1,2], Ing-Shouh Hwang [ID] [3] & See-Yeun Ting [ID] [1,2,4 ✉]

## Abstract

**Bacteria in polymicrobial habitats are constantly exposed to biotic threats from bacteriophages (or "phages"), antagonistic bacteria, and predatory eukaryotes. These antagonistic interactions play crucial roles in shaping the evolution and physiology of bacteria. To survive, bacteria have evolved mechanisms to protect themselves from such attacks, but the fitness costs of resisting one threat and rendering bacteria susceptible to others remain unappreciated. Here, we examined the fitness consequences of phage resistance in *Salmonella enterica*, revealing that phage-resistant variants exhibited significant fitness loss upon co-culture with competitor bacteria. These phage-resistant strains display varying degrees of lipopolysaccharide (LPS) deficiency and increased susceptibility to contact-dependent interbacterial antagonism, such as the type VI secretion system (T6SS). Utilizing mutational analyses and atomic force microscopy, we show that the long-modal length O-antigen of LPS serves as a protective barrier against T6SS-mediated intoxication. Notably, this competitive disadvantage can also be triggered independently by phages possessing LPS-targeting endoglycosidase in their tail spike proteins, which actively cleave the O-antigen upon infection. Our findings reveal two distinct mechanisms of phage-mediated LPS modifications that modulate interbacterial competition, shedding light on the dynamic microbial interplay within mixed populations.**

**Keywords** Phage; Lipopolysaccharide; Interbacterial Antagonism; Bacterial Immunity; Tail Spike Protein
**Subject Category** Microbiology, Virology & Host Pathogen Interaction

## Introduction

Microbial communities are frequently subjected to antagonistic interactions within their environments, predominantly from phages, bacterial competitors, and unicellular predatory eukaryotes. Phages, the most abundant biological entities on Earth, possess the capacity to devastate entire bacterial populations through lytic cycles (Brussow and Hendrix, 2002). Similarly, bacterial competitors employ a diverse array of specialized mechanisms, such as the production of antibiotics, secretion of antimicrobial peptides, deployment of toxin delivery systems, and utilization of membrane-puncturing nanomachines, all aimed at inhibiting or killing rival bacteria (Chassaing and Cascales, 2018; Cotter et al, 2013; Garcia-Bayona and Comstock, 2018; Klein et al, 2020). Moreover, unicellular eukaryotic predators, such as free-living protozoa, exert predation pressure on bacterial populations through phagocytosis (Hoque et al, 2023). Collectively, these antagonistic interactions impose significant selective pressures on bacteria that profoundly influence their evolution, ecology, and physiology.

To survive in such hostile environments, bacteria have evolved a broad spectrum of defense mechanisms against biotic threats (Smith et al, 2023). To mitigate or evade phage infection, bacteria have developed strategies targeting nearly every stage of the phage life cycle (Murtazalieva et al, 2024). Extracellular defenses, including mutations or downregulation of surface receptors (e.g., membrane proteins and lipopolysaccharides), can prevent phage attachment and subsequent infection (Kaur et al, 2021; Kulikov et al, 2019). Intracellularly, widely distributed defense systems—such as restriction-modification systems, CRISPR-Cas systems, and abortive infection mechanisms, act to neutralize phages post-entry (Doron et al, 2018; Murtazalieva et al, 2024; Vassallo et al, 2022). Parallel to these antiviral defenses, bacteria have also evolved strategies to counteract the antagonism posed by competitor bacteria. For instance, the acquisition of immunity determinants endows resistance against specific interbacterial toxins and antimicrobial compounds (Kennedy and Comstock, 2024; Peterson et al, 2020). Furthermore, general stress responses and competition-sensing pathways enhance bacterial resilience to antimicrobial agents in both environmental and clinical contexts (Cornforth and Foster, 2013; Hersch et al, 2020b; LeRoux et al, 2015). This dynamic interplay between microbial offense and defense underscores the central importance of these interactions in shaping microbial communities.

Although biotic pressures imposed by other microbes significantly drive the evolution of bacterial immunity, accumulating evidence indicates that this evolutionary process is frequently associated with fitness trade-offs (Mangalea and Duerkop, 2020). A

[1]Molecular and Cell Biology, Taiwan International Graduate Program, Academia Sinica and National Defense Medical Center, Taipei 11490, Taiwan. [2]Institute of Molecular Biology, Academia Sinica, Taipei 11529, Taiwan. [3]Institute of Physics, Academia Sinica, Taipei 115201, Taiwan. [4]Genome and Systems Biology Degree Program, National Taiwan University, Taipei 106319, Taiwan. [5]These authors contributed equally: Chia-En Tsai, Feng-Qi Wang. ✉E-mail: syting@gate.sinica.edu.tw

prominent example is bacterial surface modifications that hinder phage attachment but concurrently compromise bacterial fitness upon encountering antibiotics or host immune systems (Chan et al, 2016; Kaur et al, 2021; Levin and Bull, 2004; Yu et al, 2024b). Similarly, CRISPR-Cas immunity has been shown to impose an increased metabolic burden on host cells, which may limit its distribution across bacterial taxa (Meaden et al, 2021; Vale et al, 2015; Westra et al, 2015; Zaayman and Wheatley, 2022). Furthermore, though overproduction of the exopolysaccharide responsible for the mucoid phenotype confers partial protection against phage infection, it often incurs fitness costs in the context of bacterial competition (Chaudhry et al, 2020; Scanlan and Buckling, 2012; Wielgoss et al, 2016). Recent research on *Escherichia coli* has also demonstrated that genetic variants resistant to interbacterial intoxication may suffer from growth defects and increased susceptibility to antimicrobial agents and environmental stresses (MacGillivray et al, 2023). Such fitness trade-offs associated with antagonistic coevolution are highly context-dependent (Gomez and Buckling, 2011), but predicting their impact on microbial community composition remains challenging due to the mechanisms underlying these fitness costs frequently being poorly characterized.

In this study, we report that though bacteria develop surface modifications to protect against viral infection, these modifications result in significant trade-offs in interbacterial competition with antagonistic bacteria. We provide evidence that lipopolysaccharide (LPS), a major component of the bacterial outer membrane, plays a previously unappreciated role in protecting against contact-dependent killing by bacterial competitors. Remarkably, we have discovered that mutations in LPS biosynthesis genes, as well as interactions with phages encoding LPS-targeting endoglycosidase in their tail spike proteins, contribute to the observed competitive disadvantage. Our study identifies two distinct mechanisms by which phage-mediated LPS modifications influence interbacterial competition, highlighting the role of antagonistic coevolution in shaping bacterial composition within polymicrobial communities.

# Results

## Phage-resistant *S. enterica* displays competitive fitness deficits in co-culture with bacterial competitors

To investigate if phage-resistant bacteria incur fitness consequences when interacting with bacterial competitors, we began by obtaining phage-resistant variants of the model bacterium, *Salmonella enterica* serovar Typhimurium, from a newly generated transposon insertion library (~12,000 variants) for pairwise bacterial co-culture experiments (Trotereau et al, 2023; York et al, 1998) (Fig. 1A). We reasoned that using this model bacterium and the mutant library would facilitate the identification of underlying pathways associated with competitive fitness costs. Consequently, we isolated distinct *S. enterica* variants that were resistant to model phages from different families of Caudovirales: Felix O1 (Myoviridae) (Whichard et al, 2010), P22 (Podoviridae) (Vander Byl and Kropinski, 2000), and Chi (Siphoviridae) (Lee et al, 2013) (Fig. EV1A,B). Mutants exhibiting no significant growth defects under standard conditions were selected for further analysis (Fig. EV1C,D). We then co-cultured individual strains with competitor bacteria, i.e.,

Enterohemorrhagic *Escherichia coli* (EHEC) or *Enterobacter cloacae*, in a one-to-one ratio in both liquid and solid media. We selected EHEC and *E. cloacae* due to their ecological overlap with *S. enterica* in the mammalian gastrointestinal tract, where they frequently co-occur during infections, representing a relevant model for studying competitive interactions in both environmental and clinical contexts (Davin-Regli and Pages, 2015; Galan, 2021; Nguyen and Sperandio, 2012).

Compared to the wild-type *S. enterica* strain, we found that none of the phage-resistant mutants exhibited competitive fitness differences in liquid broth co-culture assays with EHEC or *E. cloacae* (Fig. 1B). However, in solid media, though phage-resistant mutants exhibited no observable fitness cost when incubated with EHEC, strains resistant to the Felix O1 and P22 phages displayed significant competitive disadvantage (relative survival <1) upon co-culture with *E. cloacae* (Fig. 1C). This competitive phenotype was restricted to solid media, where cell-cell contacts are relatively stable, indicating that the stress imposed by *E. cloacae* is likely contact-dependent (Vis et al, 2020). Interestingly, resistant strains against phage Chi did not present a similar fitness disadvantage, indicating that phage-specific resistance mechanisms might underlie the competitive trade-offs in *S. enterica* (Fig. 1C). Although the diversity of transposon mutants used in the initial screening was limited and may have potentially overlooked phage resistance genes, we prioritized investigating: (1) which gene mutations in *S. enterica* conferring resistance to phages Felix O1 and P22 impose interbacterial competitive disadvantages on the bacteria; and (2) what mechanism in *E. cloacae* is critical for the competitive phenotype observed in our co-culture assay.

## Mutations in LPS biosynthesis genes confer competitive disadvantages on *S. enterica*

To escape viral infection, bacteria often evolve and modify their surface receptors to prevent phage entry (Rostol and Marraffini, 2019). Previous studies have identified LPS on the outer bacterial membrane as the primary receptor for phages Felix O1 and P22 (Baxa et al, 1996; Hudson et al, 1978). Accordingly, we postulated that our phage-resistant *S. enterica* strains likely harbor mutations in LPS-related genes, thereby influencing their susceptibility to *E. cloacae*. Indeed, we identified disruptive transposon insertions concentrated in genes associated with the LPS biosynthesis pathway in our phage-resistant strains. Among these, four LPS-related genes (*waaO*, *waaJ*, *waaL*, and *wbaN*) were disrupted in the P22-resistant variants, while two (*waaO* and *dagR*) were disrupted in the Felix O1-resistant strains. These results indicate an impairment in the outer membrane cell envelope of these mutants (Fig. EV2A,B). To confirm this finding, we compared LPS levels in these mutants relative to the wild-type strain. LPS PAGE profiles revealed a significant reduction in LPS levels, particularly of the O-antigen, in the Felix O1- and P22-resistant strains compared to wild-type (Fig. 2A,B). In contrast, the phage Chi-resistant *S. enterica* variants had transposon insertions in genes unrelated to LPS biosynthesis, and their LPS levels were comparable to wild-type bacteria (Fig. EV2C,D). Additionally, many of the Felix O1- and P22-resistant strains exhibited auto-aggregation in monocultures (Fig. EV2E,F), a phenotype associated with LPS deficiency (Ge et al, 2021; Nakao et al, 2012). These findings indicate that disruption of LPS biosynthesis genes contributes to the competitive fitness defects observed in *S. enterica* co-cultured with *E. cloacae* (Fig. 1C).

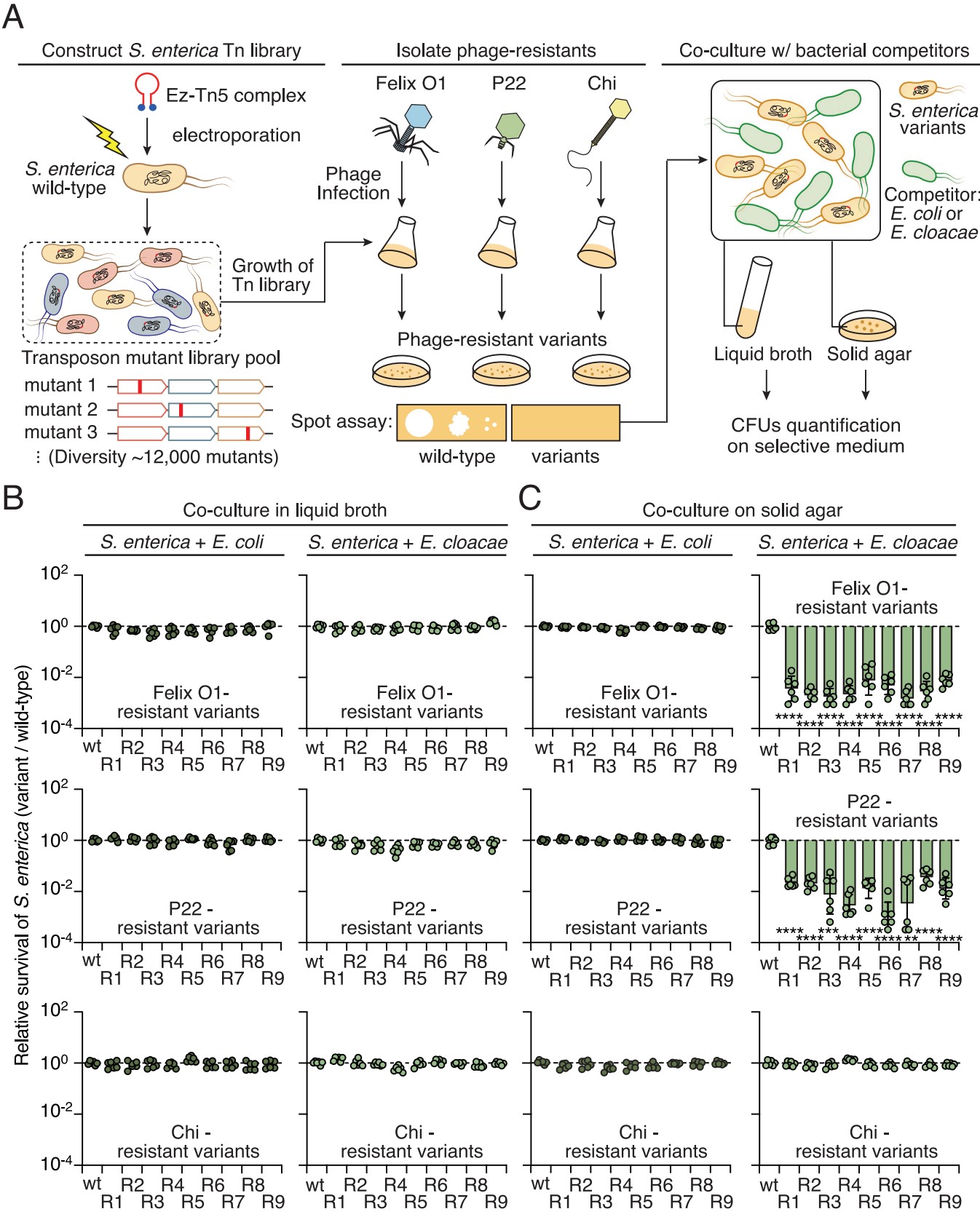

Figure 1.   Isolation of phage-resistant *S. enterica* for pairwise co-culture experiments.

(A) Schematic workflow for isolating phage-resistant strains from a transposon library (Tn-library) pool containing approximately 12,000 mutants. Detailed procedures are provided in "Methods". Phages from different families of Caudovirales were used for selection: Felix O1, P22, and Chi (Fig. EV1A). Spot assays were employed to validate the resistance of the isolated strains to phage infections (Fig. EV1B). Validated resistant variants were subsequently co-cultured with *E. coli* or *E. cloacae* in liquid broth or on solid agar. The survival of *S. enterica* was quantified by counting CFUs on selective media. (B, C) Relative survival of the indicated *S. enterica* strains compared to wild-type *S. enterica* grown in co-culture with *E. coli* or *E. cloacae* in LB broth (B) or solid agar (C). Relative survival was determined by comparing CFU ratios of phage-resistant variants and the wild-type control. Data in (B, C) are represented as means ± SD ($n = 6$). Significance was calculated based on an unpaired two-tailed Student's $t$ test. Asterisks indicate statistically significant differences between the survival of a mutant strain and the wild-type control (for P22-R7, $P = 0.00017$. **$P < 0.001$, ***$P < 0.0001$, ****$P < 0.00001$). Source data are available online for this figure.

## Phage-resistant *S. enterica* strains are more susceptible to T6SS-mediated intoxication

Next, we sought to elucidate the underlying cell contact-dependent mechanisms utilized by *E. cloacae* in competing with *S. enterica*. Prior studies have shown that *E. cloacae* relies on functional secretory systems, such as the type VI secretion system (T6SS) and the contact-dependent growth inhibition (CDI) pathway, to outcompete their bacterial rivals (Beck et al, 2014; Whitney et al, 2014). These secretory systems translocate antibacterial toxins into neighboring cells upon direct cell-cell interaction, facilitating potent mechanisms of contact-dependent intoxication between bacteria (Aoki et al, 2005; Hood et al, 2010). Therefore, we hypothesized that the competitive fitness trade-offs observed in the phage-resistant strains might be attributable to interbacterial antagonism mediated by these secretory systems.

To determine if such antagonistic pathways in *E. cloacae* contribute to the competitive defects of the Felix O1- and P22-resistant *S. enterica* strains, we subjected them to pairwise competition under contact-promoting conditions against *E. cloacae* mutant strains in which the T6SS-1 (ΔtssM) or CDI (ΔcdiA) had been inactivated (Beck et al, 2014; Whitney et al, 2014). Consistent with our co-culture experiments, the phage-resistant variants displayed reduced competitiveness when incubated with wild-type *E. cloacae* (Fig. 2C,D). Notably, T6SS inactivation abrogated the competitive defect of the phage-resistant strains, whereas the competitive phenotype persisted in competition with *E. cloacae* ΔcdiA (Fig. 2E–H). These results support the notion that the competitive fitness deficit in the phage-resistant strains arises from T6SS-mediated interbacterial antagonism.

## LPS O-antigen protects *S. enterica* from T6SS-mediated assaults

Although the T6SS is an effective antibacterial mechanism, target cells can evade being killed by avoiding direct contact with bacteria possessing an active T6SS (Hersch et al, 2020a). One strategy to do so is for target cells to produce extracellular biopolymers, such as secreted exopolysaccharides, membrane-attached capsules, or colanic acid, all of which may act as a physical barrier against T6SS-mediated attacks (Flaugnatti et al, 2021; Hersch et al, 2020b; Toska et al, 2018). LPS, a major component of Gram-negative bacterial surfaces, is known to confer protection against external stresses such as antibiotics, antimicrobial peptides, and host immune systems (Papo and Shai, 2005; West et al, 2005). Therefore, the competitive disadvantage we observed in LPS-deficient *S. enterica* led us to explore the function of LPS in protecting against T6SS-mediated assaults.

LPS is comprised of lipid A covalently linked to core oligosaccharides (core OS) and repetitive O-units (O-antigen) (Fig. 3A). First, we sought to determine if the sugar composition of the core OS is involved in protecting *S. enterica* against T6SS. We constructed in-frame deletions of LPS-related genes in *S. enterica* differing in core OS structures (i.e., WaaL, WaaJ, WaaO, WaaG for the main chain OS structure, or WaaK, WaaB, WaaQ, WaaY for the side chain) (Bertani and Ruiz, 2018) (Fig. 3A). If the LPS structure exerts a protective role, we anticipated that mutants hosting these mutated genes would exhibit competitive fitness defects upon co-incubation with *E. cloacae* carrying an active T6SS. Indeed, though mutations in the LPS-related genes did not result in significant growth defects in monocultures (Fig. EV3A–C), strains deficient in the genes essential for building the main chain OS structure displayed lower interbacterial competitiveness than the wild-type strain (Figs. 3B,C and EV3D), similar to the results we observed for LPS-deficient phage-resistant variants (Fig. 2C,D). In contrast, *S. enterica* mutants in which the side chain OS structure was perturbed presented indistinguishable competitiveness to wild-type, except for the WaaK-deficient strain (Figs. 3D,E and EV3E). Previous studies demonstrated that WaaK is required for proper ligation of O-antigen to the core OS of LPS (Kaniuk et al, 2004). Indeed, LPS PAGE profiles support the finding that O-antigen levels in the ΔwaaK strain were reduced significantly, i.e., to levels similar to those in the O-antigen ligase-knockout cells (ΔwaaL) (Fig. 3E). Together, these results indicate that the O-antigen extending from the core OS plays a crucial role in protecting *S. enterica* from T6SS-mediated interbacterial antagonism.

O-antigens are hypervariable repeat polysaccharides that present heterogeneity and a wide range of chain lengths, which determine the uniqueness of bacterial serotypes (Di Lorenzo et al, 2022). Previous studies have demonstrated that chain length regulators, such as WzzB and FepE, determine O-antigen length and degree of polymerization in *S. enterica* (Bastin et al, 1993; Murray et al, 2003). Our LPS PAGE profiles indicate that WzzB is responsible for producing long (L)-modal length O-antigen chains (16–35 sugar repeat units), whereas FepE is responsible for very long (VL) ones (>100 sugar repeat units) (Fig. 3F). To better define the protective role of O-antigen in protecting against interbacterial antagonism, we constructed ΔwzzB and ΔfepE strains and subjected them to competition assays against *E. cloacae* carrying an active T6SS. Interestingly, only the ΔwzzB cells displayed significant competitive defects, whereas ΔfepE cells showed wild-type levels of protection (Fig. 3G). In addition, both mutants showed comparable survival in competition with *E. cloacae* ΔtssM (Fig. EV3F). Therefore, our mutational analyses and LPS PAGE profiles indicate that L-modal O-antigen is necessary to protect *S. enterica* from T6SS intoxication.

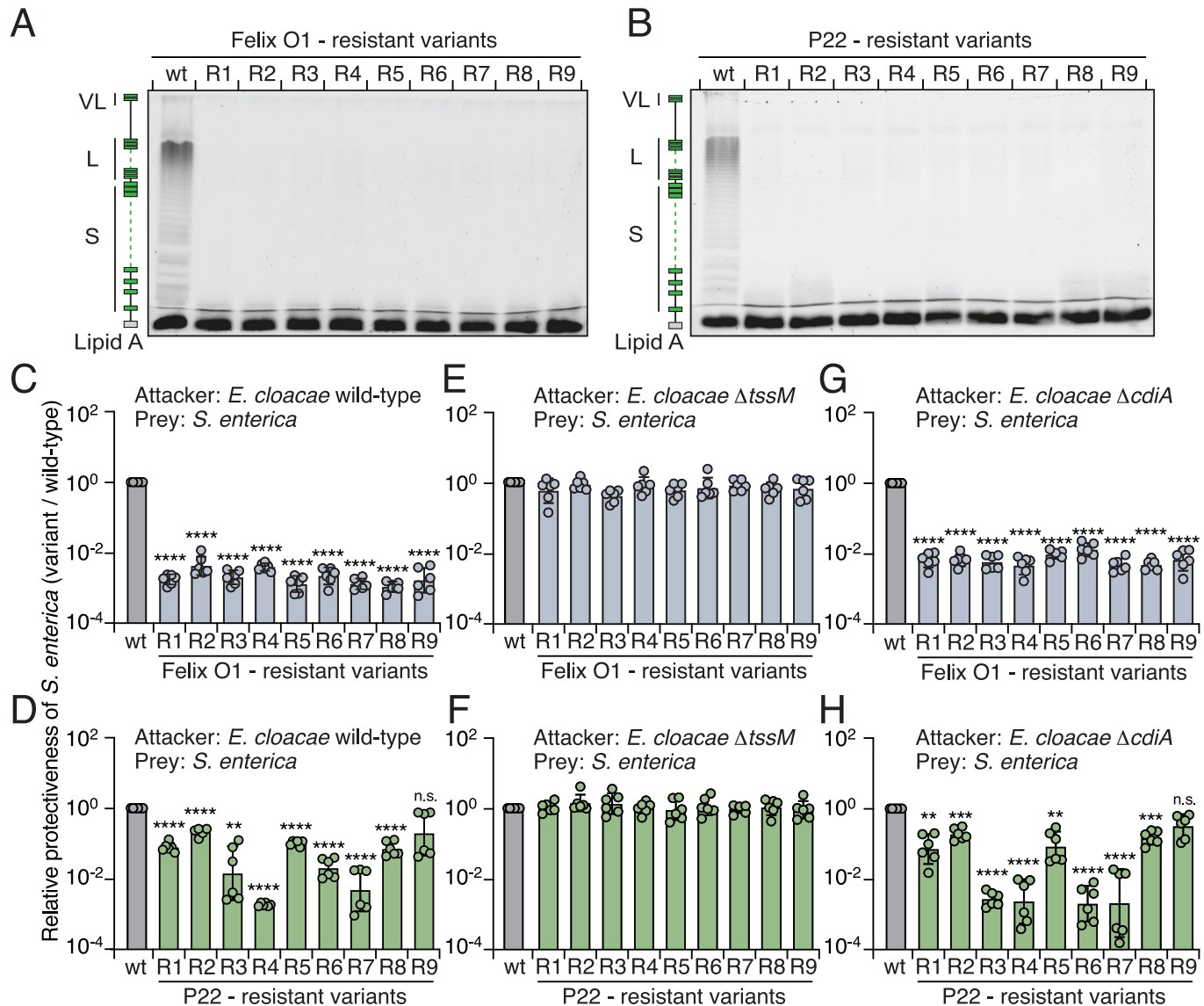

**Figure 2. Phage-resistant strains exhibit LPS deficiency and increased susceptibility to T6SS-mediated antagonism by *E. cloacae*.**

(A, B) 12% LPS PAGE profiles showing the levels of LPS produced by the indicated *S. enterica* strains: (A) LPS from the Felix O1-resistant strains; (B) LPS from the P22-resistant strains. A schematic representation of LPS is shown on the left. (C–H) Protectiveness of the indicated *S. enterica* phage-resistant strains relative to the wild-type strain after competition with *E. cloacae* wild-type (C, D), Δ*tssM* (E, F), and Δ*cdiA* (G, H) strains. Data in (C–H) are presented as means ± SD (*n* = 6). Significance was calculated based on an unpaired two-tailed Student's *t* test. Asterisks indicate statistically significant differences between the relative protective index of a mutant strain and the wild-type control (D, *P* = 0.0004 for P22-R3. (H) *P* = 0.0002 for P22-R1, and *P* = 0.0004 for P22-R5. **$P < 0.001$, ***$P < 0.0001$, ****$P < 0.00001$, and ns not statistically significant). Source data are available online for this figure.

However, our data do not exclude the possibility that VL-modal O-antigen in *S. enterica* may also offer protection against T6SS-mediated attacks by *E. cloacae*. In wild-type *S. enterica*, the VL-modal O-antigen is present at significantly lower levels than the L-modal O-antigen (Fig. 3F). We hypothesized that increasing the abundance of VL-modal O-antigen could enhance the protective ability during interbacterial competition. To test this, we heterologously overexpressed *fepE* in wild-type *S. enterica*, leading to elevated levels of VL-modal O-antigen (Murray et al, 2006) (Fig. 3H). Notably, we observed that cells with overproduction of VL-modal length O-antigen exhibited enhanced protection during interbacterial competition compared to control cells harboring an empty vector (Fig. 3I). These finding suggest that in wild-type *S. enterica*, L-modal O-antigen is sufficient to confer protection against T6SS intoxication. However, when overproduced, VL-modal O-antigen can also play a protective role during interbacterial competition.

## LPS-mediated protection against T6SS attack extends to other Gram-negative bacteria

Though LPS structure varies among Gram-negative species (Di Lorenzo et al, 2022; Zhang et al, 2013), we speculated that its intrinsic protective capacity against T6SS could be broadly

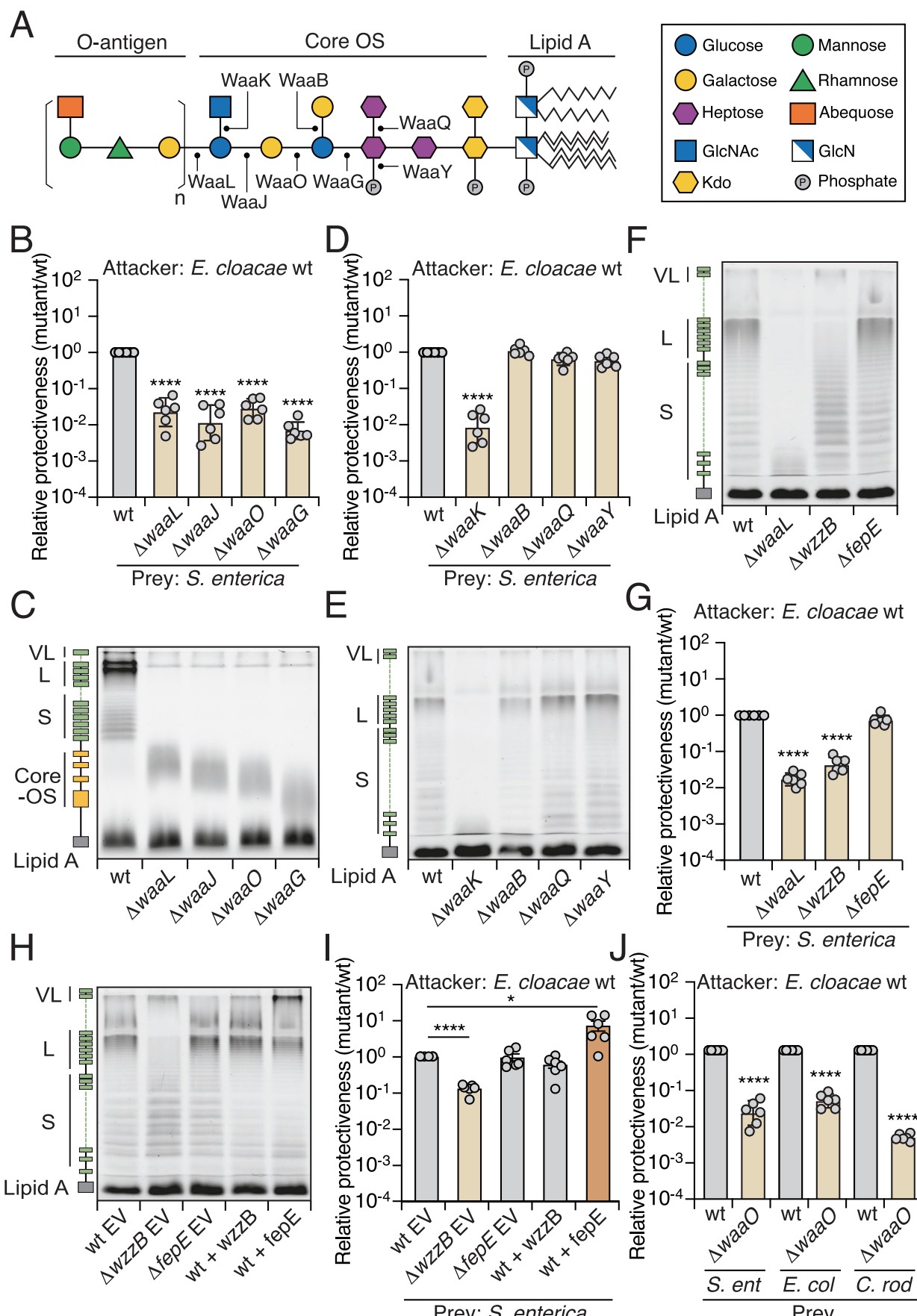

**Figure 3. LPS protects bacteria from T6SS-mediated assaults.**

(A) Schematic representation of the LPS structure in *S. enterica*. Glycosyltransferases involved in LPS biosynthesis are indicated. (B, D, G, I) Protectiveness of the indicated *S. enterica* mutant or indicated protein overexpressing strains relative to the wild-type strain after competition with wild-type *E. cloacae*. (C, E, F, H) LPS PAGE profiles: (C) 20% gel; (E, F, H) 12% gel, showing LPS levels produced by the indicated *S. enterica* mutant or indicated protein overexpressing strains. EV empty vector. (J) Protectiveness of the indicated bacterial species (Δ*waaO* mutant) relative to the wild-type strain after competition with wild-type *E. cloacae*. Data in (B, D, G, I, J) are presented as means ± SD (*n* = 6). Significance was calculated based on an unpaired two-tailed Student's *t* test. Asterisks indicate statistically significant differences between the relative protective index of a mutant strain and the wild-type control (I, *P* = 0.008 for WT+fepE. *P* < 0.05, ****P* < 0.00001). Source data are available online for this figure.

conserved. We tested this hypothesis by examining the protective function of LPS from two distantly related model species, Enterohemorrhagic *E. coli* (EHEC) and *Citrobacter rodentium*. We constructed LPS-deficient strains (Δ*waaO*) of these species and then subjected them to pairwise competition assays against *E. cloacae* with or without an active T6SS. Compared to the wild-type strain, LPS-deficient cells presented significant competitive deficits when incubated with *E. cloacae* carrying a functional T6SS (Figs. 3J and EV4A). Inactivation of the T6SS abrogated the fitness defects of those variants in the competition assay, supporting the notion that LPS-mediated protection against the T6SS is conserved beyond *S. enterica* (Fig. EV4B).

Next, we sought to determine if LPS also provide protection against T6SS-mediated attacks from other antagonistic bacteria that secrete distinct sets of T6SS effectors. For this purpose, we employed *Burkholderia thailandensis*, a species with a well-characterized T6SS, and conducted competition assays with wild-type *S. enterica* and its LPS-deficient variant (Schwarz et al, 2010). As expected, the LPS-deficient strain displayed reduced protection when co-cultured with wild-type *B. thailandensis*, but not with the Δ*tssM* (Δ*BTH_I2954*) strain (Fig. EV4C,D). These results suggest that the protective role of LPS against T6SS-mediated antagonism is widespread across bacterial species and effective against diverse T6SS-bearing competitors.

### LPS-targeting endoglycosidase in phage tail spike protein increases susceptibility to interbacterial antagonism

We have demonstrated herein that mutations in *S. enterica* genes involved in LPS biosynthesis lead to increased susceptibility to T6SS-mediated attacks. Nevertheless, apart from genetic variations in bacteria altering LPS structures, external stresses can also induce LPS modifications, thereby compromising their protective capacity against antimicrobials from predators and hosts, as well as environmental factors (Simpson and Trent, 2019). Consequently, we wanted to examine if external stress-mediated LPS modification could influence the susceptibility of *S. enterica* during interbacterial competition.

In our investigation into potential external modifiers of LPS, we found that many phages, including our model phage P22, encode a dual-function tail spike protein (TSP) with LPS-targeting endoglycosidase activity (Steinbacher et al, 1997) (Figs. 4A and EV1A). This phage protein mediates recognition of the bacterial host LPS receptor and hydrolyzes the O-antigen, allowing phages to penetrate the bacterial cell envelope and inject their DNA upon viral infection (Baxa et al, 1996; Eriksson et al, 1979) (Fig. 4B,C). Notably, the TSP is highly active and stable under extreme conditions, making it an ideal external agent to test our hypothesis

without genetically manipulating *S. enterica* (Steinbacher et al, 1994).

Consistent with previous reports, the TSP from phage P22 could be heterologously expressed and affinity-purified in high amounts (Waseh et al, 2010) (Appendix Fig. S1A). LPS PAGE profiles indicated that wild-type TSP, but not the catalytic aspartate mutant TSP$_{D392N}$, degraded the O-antigen of *S. enterica* (Fig. 4D). Cells treated with TSP (wild-type or D392N) did not exhibit detectable growth defects in monocultures (Appendix Fig. S1B), but phage titers were dramatically reduced, indicating that TSP treatment prevents P22 from recognizing the LPS receptor (Fig. 4E). Next, we assessed if TSP-treated *S. enterica* exhibits increased susceptibility to T6SS-mediated assaults. Indeed, TSP-treated cells showed reduced interbacterial competitiveness when co-incubated with *E. cloacae* possessing an active T6SS compared to the control experiment using either wild-type cells or cells treated with TSP$_{D392N}$ (Fig. 4F). The resulting competitive fitness defect was similar to that observed for the LPS-deficient strain (Δ*waaL*) (Fig. 4F). Inactivation of the T6SS abrogated the fitness defects in TSP-treated cells (Fig. 4G). These results indicate that external LPS modifiers from phages, such as TSP with endoglycosidase activity, can sensitize *S. enterica* to T6SS-mediated attacks.

### Phage-hosted endoglycosidase sensitizes *S. enterica* to interbacterial antagonism in mixed populations

Our data to this point support a model whereby the endoglycosidase activity of TSP cleaves LPS O-antigen on bacterial cell surfaces, thereby sensitizing the bacteria to T6SS-mediated assaults (Fig. 4F,G). However, treating cells with concentrated, purified TSP might yield non-physiologically relevant outcomes. In natural polymicrobial communities, TSP is anchored to the phage tail baseplate during viral particle assembly (Tang et al, 2011), so the free form of TSP is likely present at much lower concentrations than those used in our assays. Therefore, we investigated if the presence of intact phage virions containing enzymatically active TSP could sensitize target bacteria to interbacterial antagonism (Fig. 5A).

Remarkably, when we incubated *S. enterica* with both phage P22 and *E. cloacae* carrying an active T6SS, *S. enterica* survival significantly decreased (Fig. 5B). The deficient fitness phenotype was abolished when the T6SS of *E. cloacae* was inactivated or when *S. enterica* was incubated with Felix O1 or Chi phages, which lack endoglycosidase activity in their tail spikes or tail fibers (Sonani et al, 2024; Whichard et al, 2010) (Fig. 5B,C). These results corroborate our hypothesis that phages containing an endoglycosidase targeting LPS sensitize host bacteria to T6SS-mediated attacks in mixed microbial populations.

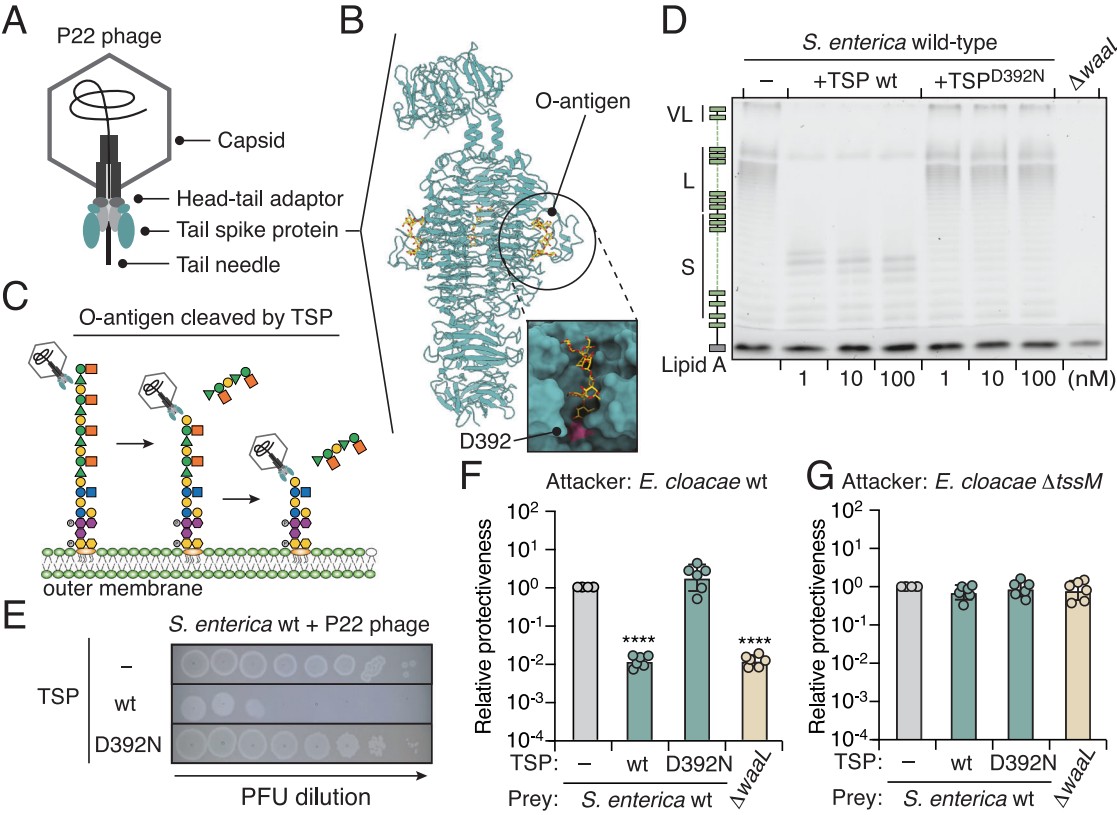

**Figure 4. Purified TSP increases the susceptibility of *S. enterica* to T6SS-mediated intoxication.**

(A) Schematic illustration of phage P22. (B) Crystal structure of the P22 TSP (PBD: 1TYX) (Steinbacher et al, 1996) with the O-antigen substrate bound in the catalytic pocket. The D392 residue is highlighted in pink. (C) Schematic representation of O-antigen digestion by phage P22 hosting an enzymatically active TSP. (D) 12% LPS PAGE profiles showing LPS levels from wild-type *S. enterica* treated with different concentrations of purified TSP (wild-type or D392N mutant) for 10 min at room temperature. LPS extracted from the Δ*waaL* strain is shown as a control. (E) Tenfold serial dilution spot assay of P22 phage on wild-type *S. enterica* treated with 10 nM purified TSP. The image is representative of triplicate experiments. (F, G) Protectiveness of the wild-type *S. enterica* strain treated with purified TSP (wild-type or D392N mutant) relative to the mock-treated *S. enterica* after competition with *E. cloacae* wild-type (F) or Δ*tssM* (G) strains. Data in (F, G) are presented as means ± SD (*n* = 6). Significance was calculated based on an unpaired two-tailed Student's *t* test. Asterisks indicate statistically significant differences between the relative protective index of TSP-treated *S. enterica* and the mock-treated control (****$P < 0.00001$). Source data are available online for this figure.

To validate this finding, we first ensured that *S. enterica* did not develop resistance to P22 during the phage-bacterium co-culture assay. Reinfecting surviving *S. enterica* with P22 resulted in plaque titers comparable to those before the experiment, indicating that the competitive phenotype was unlikely due to genetic adaptation (Appendix Fig. S2A). Next, we confirmed that P22 does not cross-infect *E. cloacae*, as revealed by no P22 plaques being observed even at a high plaque-forming unit (PFU) concentration using *E. cloacae* as the host (Appendix Fig. S2B). To further minimize the impact of viral infection from different phages, we subjected phages to 254 nm UV-C irradiation to crosslink their DNA, preventing DNA injection into *S. enterica* while retaining the enzymatic activity of TSP in the tail spikes (Fig. 5D). Strikingly, though phage titers decreased dramatically following UV treatment, the competitive fitness deficits against the T6SS persisted in assays with *E. cloacae* and UV-treated P22 (Fig. 5E–G). Again, no detectable competitive fitness differences were observed when *S. enterica* was incubated with UV-treated Felix O1 or Chi (Fig. 5F,G; Appendix Fig. S2C,D). Together, these outcomes indicate that phages with endoglycosidase activity in their TSPs for binding and hydrolyzing LPS increase the susceptibility of *S. enterica* to interbacterial antagonism in mixed microbial populations.

## Newly Isolated environmental phages that sensitize *S. enterica* to interbacterial antagonism

We next sought to determine if the ability to sensitize *S. enterica* to interbacterial antagonism is specific to the model phage P22 or if this trait is widespread in natural environments. To investigate this, we isolated phages targeting *S. enterica* from soil samples collected in Northern Taiwan. Four phages, designated As1 through As4, were successfully identified, each exhibiting unique shapes and plaque morphologies (Fig. 5H; Appendix Fig. S2E). These newly isolated phages were then assessed for their impact on interbacterial competition.

Notably, two of the four phages, As1 and As3, both belonging to the family Podoviridae, significantly reduced the competitiveness of *S. enterica* when co-incubated with *E. cloacae* possessing an active T6SS, similar to the effect observed with phage P22 (Fig. 5I). In contrast, co-incubation with phages As2 or As4 did not affect the competitiveness of *S. enterica* (Fig. 5I). The competitiveness of *S. enterica* remained consistent across all groups when incubating with *E. cloacae* lacked a functional T6SS (Fig. 5J). In addition, whole-genome sequencing of the newly isolated phages As1 and

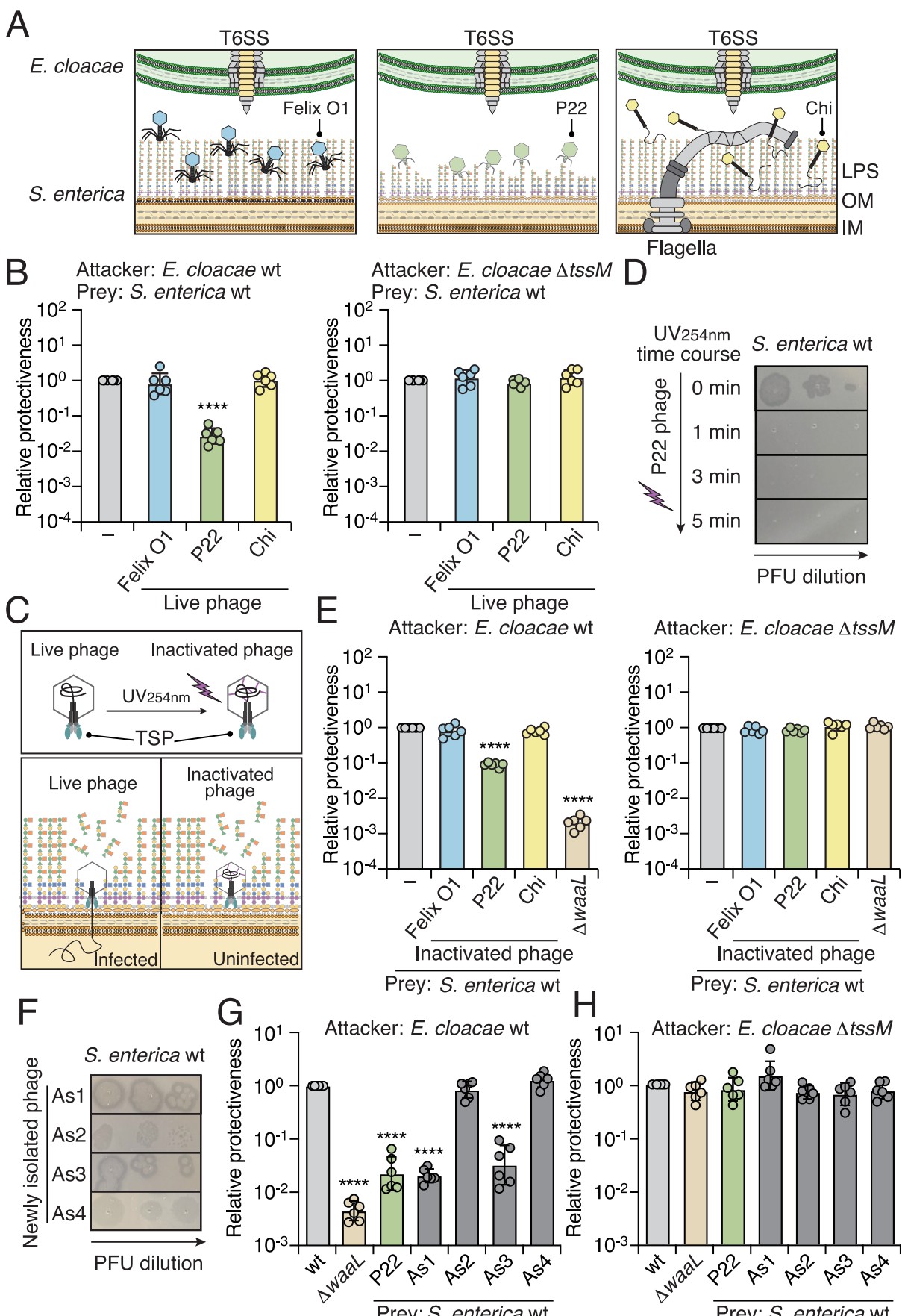

**Figure 5.  Phage P22 sensitizes *S. enterica* to T6SS-mediated attacks in mixed populations.**

(A) Schematic representation of phages with or without LPS-targeting endoglycosidase during infection of *S. enterica*. *E. cloacae* carrying a functional T6SS is depicted at the top. (B) Protectiveness of the wild-type *S. enterica* strain treated with different phages (Felix O1, P22, or Chi) relative to the mock-treated *S. enterica* after competition with *E. cloacae* wild-type or Δ*tssM* strains. (C) Schematic of P22 with UV$_{254nm}$ treatment. (D) Tenfold serial dilution spot assay of UV-inactivated P22 phage on wild-type *S. enterica*. The image is representative of triplicate experiments. (E) Protectiveness of the wild-type *S. enterica* strain treated with UV-inactivated phage particles (Felix O1, P22, or Chi) relative to the mock-treated *S. enterica* after competition with *E. cloacae* wild-type or Δ*tssM* strains. (F) Tenfold serial dilution spot assay of phages As1, As2, As3, and As4. (G, H) Protectiveness of the wild-type *S. enterica* strain treated with different phages (P22, As1, As2, As3, or As4) relative to the mock-treated *S. enterica* after competition with *E. cloacae* wild-type (G) or Δ*tssM* (H) strains. Data in (B, E, G, H) are presented as means ± SD ($n = 6$). Significance was calculated based on an unpaired two-tailed Student's *t* test. Asterisks indicate statistically significant differences between the relative protective index of phage-treated or UV-inactivated phage-treated *S. enterica* and the mock-treated control (****$P < 0.00001$). Source data are available online for this figure.

As3 revealed the presence of *tsp* genes encoding predicted endoglycosidases. These findings suggest that the ability of phages to sensitize bacteria to interbacterial antagonism is prevalent in natural environments.

## Evidence that the O-antigen of LPS serves as a physical barrier against interacting bacteria

The LPS-containing outer membrane of Gram-negative species serves as a barrier against various harmful agents, including hydrophobic antibiotics, antimicrobial peptides, detergents, and bile acids (Delcour, 2009; Snyder and McIntosh, 2000). The structural complexity of LPS is critical for regulating membrane permeability, which may be compromised by any modifications (Zgurskaya et al, 2015). Previous studies have demonstrated that LPS remodeling can alter the negative charge on the bacterial surface or affect the binding of divalent cations, thereby impacting the stability of the outer membrane (Vaara, 1992). Therefore, one plausible explanation for the observed interbacterial competitive defects displayed by our LPS O-antigen-deficient variants is an increase in the permeability of their outer membrane. To investigate this possibility, we performed two standard outer membrane permeability assays: (i) N-phenyl-1-napthylamine (NPN) uptake; and (ii) vancomycin (VAN) sensitivity. NPN is a hydrophobic fluorescent probe that exhibits increased fluorescence upon interacting with the hydrophobic core of a disrupted membrane, making it a reliable indicator of membrane permeability (Helander and Mattila-Sandholm, 2000). Vancomycin, a glycopeptide antibiotic, is typically ineffective against Gram-negative bacteria due to its large molecular size and inability to penetrate an intact outer membrane. However, its efficacy can increase if membrane integrity is compromised (Bolla et al, 2011). Therefore, these assays can provide robust measures of outer membrane permeability.

In the NPN uptake assay, we observed that wild-type cells pretreated with the metal chelator ethylenediaminetetraacetic acid (EDTA) showed a significant increase in NPN fluorescence, indicative of membrane disruption. However, no significant difference in NPN uptake was observed between the wild-type cells and the O-antigen-deficient mutants (i.e., Δ*wzzB*, which lacks L-modal length O-antigen, and Δ*waaL* that lacks all O-antigens) (Fig. 6A). Similarly, in the VAN sensitivity assay, neither mutant strain demonstrated increased sensitivity to vancomycin compared to the wild-type strain (Fig. 6B). These results are in line with previous studies on LPS in *E. coli*, which have shown that the core OS of LPS is particularly crucial for maintaining outer membrane permeability (Wang et al, 2015). As both the Δ*wzzB* and Δ*waaL*

mutants preserve the core OS structure, their increased susceptibility to T6SS-mediated attacks is unlikely to be attributable to compromised outer membrane permeability.

Given that the T6SS punctures and injects toxins into interacting bacteria in a contact-dependent manner (Allsopp and Bernal, 2023), we next hypothesized that the O-antigen of LPS might act as a physical barrier, partially hindering T6SS-mediated penetration by limiting close-range interaction. To test this hypothesis, we immobilized *S. enterica* wild-type and O-antigen-deficient mutants on a mica substrate and employed atomic force microscopy (AFM) to characterize bacterial surface morphology (Dufrene, 2002) (Fig. 6C). Interestingly, the resolved AFM images reveal distinct and clear grooved structures on the surfaces of both the wild-type and Δ*wzzB* mutant (Fig. 6D,E). In contrast, the Δ*waaL* mutant, which retains only the core OS, displayed a more flattened surface (Fig. 6F). A similar groove-like surface morphology has been observed in other bacteria (Kahli et al, 2022; Strauss et al, 2009), and flattened cell surfaces have been noted in O-antigen-deficient cells (Pagnout et al, 2019).

To gain more insight into the mechanical properties of LPS, we further analyzed the force-distance curves, which represent the interaction between the AFM tip and the bacterial surface (Fig. 6G). The force-distance profiles of both wild-type and mutant strains exhibited two distinct force-slope changes, indicative of interactions with two different substrates (Fig. EV5). The first substrate corresponds to the LPS layer, and the second stiffer substrate represents the outer membrane and cell wall (Fig. 6C). For a more precise description, we defined point 1 as the initial contact between the AFM tip and the LPS layer ($x = 0$, $y = 0$), and point 2 as the transition point between the LPS layer and the outer membrane (Fig. 6G). By measuring the distance between points 1 and 2, we determined that LPS length in wild-type cells is $5.39 \pm 1.16$ nm, consistent with predictions for intact LPS structures (Gao et al, 2023) (Fig. 6H). In contrast, Δ*wzzB* cells, which lack L-modal O-antigen, exhibited a reduced LPS length of $3.96 \pm 0.424$ nm, and Δ*waaL* cells that retain only the core OS had an LPS length of $3.39 \pm 0.41$ nm (Fig. 6I,J). LPS length in wild-type cells was significantly more variable than in the mutants, implying heterogeneity in LPS structure (Fig. 6K). The loss of L-modal O-antigen in Δ*wzzB* led to a marked reduction in the force region, similar to the reduction observed in the Δ*waaL* mutant (Fig. 6K). These LPS lengths are comparable to those reported in studies of *E. coli* strains with the core OS but lacking O-antigen, for which LPS length was approximately $3 \pm 2$ nm (Strauss et al, 2009). The reduction in LPS length we determined for our O-antigen-deficient mutants indicates that the O-antigen functions as a physical barrier, potentially hindering close interactions with competing bacteria.

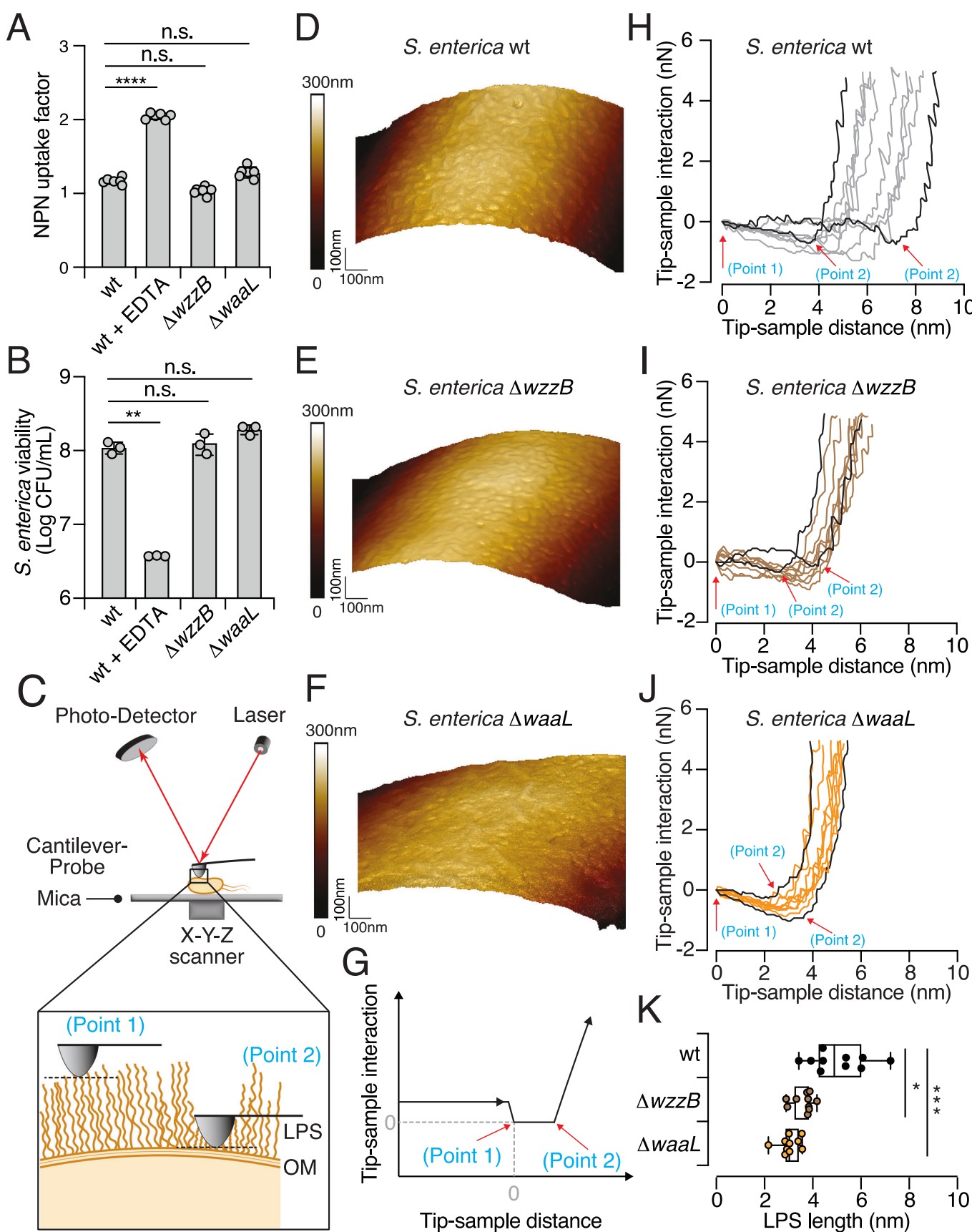

◄ **Figure 6. Evidence that the O-antigen of LPS serves as a physical barrier.**

(A, B) Measurements of outer membrane permeability of the *S. enterica* wild-type, Δ*wzzB*, and Δ*waaL* strains. EDTA (1 mM) was added to the wild-type as the control. (A) NPN uptake assay; (B) VAN sensitivity assay. (C) Schematic representation of the atomic force microscopy (AFM) setup. *S. enterica* cells were immobilized on freshly prepared mica substrates. A laser beam reflected on the back of the cantilever is used to measure the force interactions between the probe and sample. The feedback control system between the photodetector and the X–Y–Z scanner enables precise detection of the surface morphology of the cells. A zoomed-in view of AFM measurement of the bacterial surface is also shown. (D–F) Images of AFM height for the *S. enterica* wild-type (D), Δ*wzzB* (E), and Δ*waaL* (F) strains, as obtained using the PeakForce Tapping (PFT) mode. The image scan size is 1 μm × 1 μm. (G) Schematic of the AFM force-distance curve obtained from *S. enterica* samples using the Force Volume (FV) mapping mode. The point at which the tip contacts the sample is defined as zero (point 1), and slope changes indicate a transition to a stiffer substrate (point 2). (H–J) Force-distance curves for the *S. enterica* wild-type (H), Δ*wzzB* (I), and Δ*waaL* (J) strains. Ten curves were captured from each sample at randomly selected locations, with the longest and shortest LPS lengths highlighted. Red arrows mark points 1 and 2. (K) LPS length of the *S. enterica* wild-type, Δ*wzzB*, and Δ*waaL* strains was determined from force-distance curves by measuring the distance between point 1 and point 2. Data are shown as box with whiskers (*n* = 10). The box is determined by the 25 and 75 percentiles, and the whiskers are determined by min and max; the line in the box indicates the median. Data in (A, B) are presented as means ± SD (*n* = 6 in (A), and *n* = 3 in (B)). Significance was calculated based on an unpaired two-tailed Student's *t* test. Asterisks indicate statistically significant differences between wild-type strain and each mutant (B, *P* = 0.0007 for WT + EDTA. (K) *P* = 0.0017 for Δ*wzzB*. *\**P* < 0.05, **\**P* < 0.001, ***\**P* < 0.0001, ****\**P* < 0.00001 and ns not statistically significant). Source data are available online for this figure.

## Discussion

Bacterial integrity has been shaped over billions of years through interactions with other microorganisms, with antagonistic interactions serving as a key driver of bacterial evolution and physiology (Smith et al, 2023). Extensive research has elucidated the diverse mechanisms bacteria employ to defend themselves from biotic threats, highlighting their importance in the survival and adaptation of bacterial species. Although these defensive adaptations offer clear evolutionary benefits, they are not without fitness costs (Mangalea and Duerkop, 2020; Zaayman and Wheatley, 2022). Our results demonstrate that though modification of surface receptors enables bacterial populations to effectively resist and adapt rapidly to phage infection, such adaptation imposes significant fitness costs, increasing susceptibility to contact-dependent killing by other antagonistic bacteria. These findings emphasize the fine balance between adaptation and vulnerability in microbial interactions, suggesting that antagonistic coevolution plays a pivotal role in shaping the dynamics of bacterial communities.

Our results reveal a hitherto unappreciated dual role for LPS as both a receptor for phage adsorption and a protective barrier against close-contact bacterial competitors. Our observation that T6SS-mediated interbacterial antagonism is more effective against LPS-deficient strains implies a significant evolutionary trade-off, wherein resistance to phage predation may increase susceptibility to other biotic threats. Furthermore, our results indicate that the protective function of LPS against the T6SS is likely conserved across different Gram-negative bacteria, as evidenced by the similar competitive fitness defects observed in LPS-deficient variants of *E. coli* and *C. rodentium*. This evolutionary conservation points to broader ecological implications, with modifications of the LPS potentially significantly influencing microbial community composition across different populations.

The extent to which LPS protects against other contact-dependent killing mechanisms beyond the T6SS remains unclear. For example, bacterial genera such as *Xanthomonas* and *Stenotrophomonas* possess a Type IV Secretion System (T4SS) specialized in transferring toxic effectors into neighboring cells in a contact-dependent manner, leading to cell death (Bayer-Santos et al, 2019; Sgro et al, 2019; Souza et al, 2015). The evidence presented in our study supports that LPS functions as a physical barrier, potentially hindering close contact with interacting bacteria. Therefore, it is plausible that LPS might similarly interfere with toxin delivery by

other bacterial nanomachines, representing a broader defensive strategy against diverse biotic threats.

To fully understand how LPS influences T6SS effector delivery, future studies will need to address key mechanistic questions regarding the T6SS injection process. For example: (1) how deeply does the T6SS apparatus penetrate the target Gram-negative cells during injection; (2) what is the magnitude of the injection force generated by the T6SS; and (3) does the structural integrity of the T6SS apparatus remain intact throughout and after contraction? Although it has been well documented that some T6SS effectors act in the cytosol of target cells, there is evidence to suggest that cytosolic effectors are delivered initially into the periplasm and subsequently translocated into the cytosol for intoxication (Ali et al, 2023; Whitney et al, 2015). Furthermore, although contraction of the T6SS apparatus occurs within milliseconds (Basler et al, 2012; LeRoux et al, 2012), this rapid action does not preclude the possibility that the injection force could be influenced by the thickness of the LPS layer. In addition, the stability of T6SS structural or delivered proteins—such as PAAR, VgrG, and Hcp—within the delivery complex might be compromised upon encountering physical barriers such as the LPS layer and the outer membrane of target cells. These potential interactions could affect the efficiency of effector delivery, leading to reduced competitiveness during interbacterial antagonism, as shown in our study.

Interestingly, our study has also revealed that phages can influence interbacterial competition in ways beyond direct lysis. Specifically, phages with endoglycosidase activity in their TSPs can degrade the O-antigen of target bacteria, thereby sensitizing them to interbacterial antagonism. This scenario adds a new dimension to our understanding of phage-bacterial interactions, whereby phages not only act as predators but also modulate bacterial interactions by compromising the structural integrity of bacterial membranes. Notably, purified TSP, unlike whole phage particles, do not promote resistant mutations as they do not directly kill bacteria (Wang et al, 2024) (Appendix Fig. S1B). Our discovery reveals a plausible mechanism by which TSP facilitates the clearance of target cells, with antagonistic bacteria that occupy the same ecological niches potentially eliminating competitors more effectively when the LPS of those competitors is cleaved by exogenous TSP (Waseh et al, 2010).

We propose a working model of how phages may act as a key modulator of interbacterial competition within bacterial communities (Fig. 7A–F). In the absence of phages, bacterial competitors

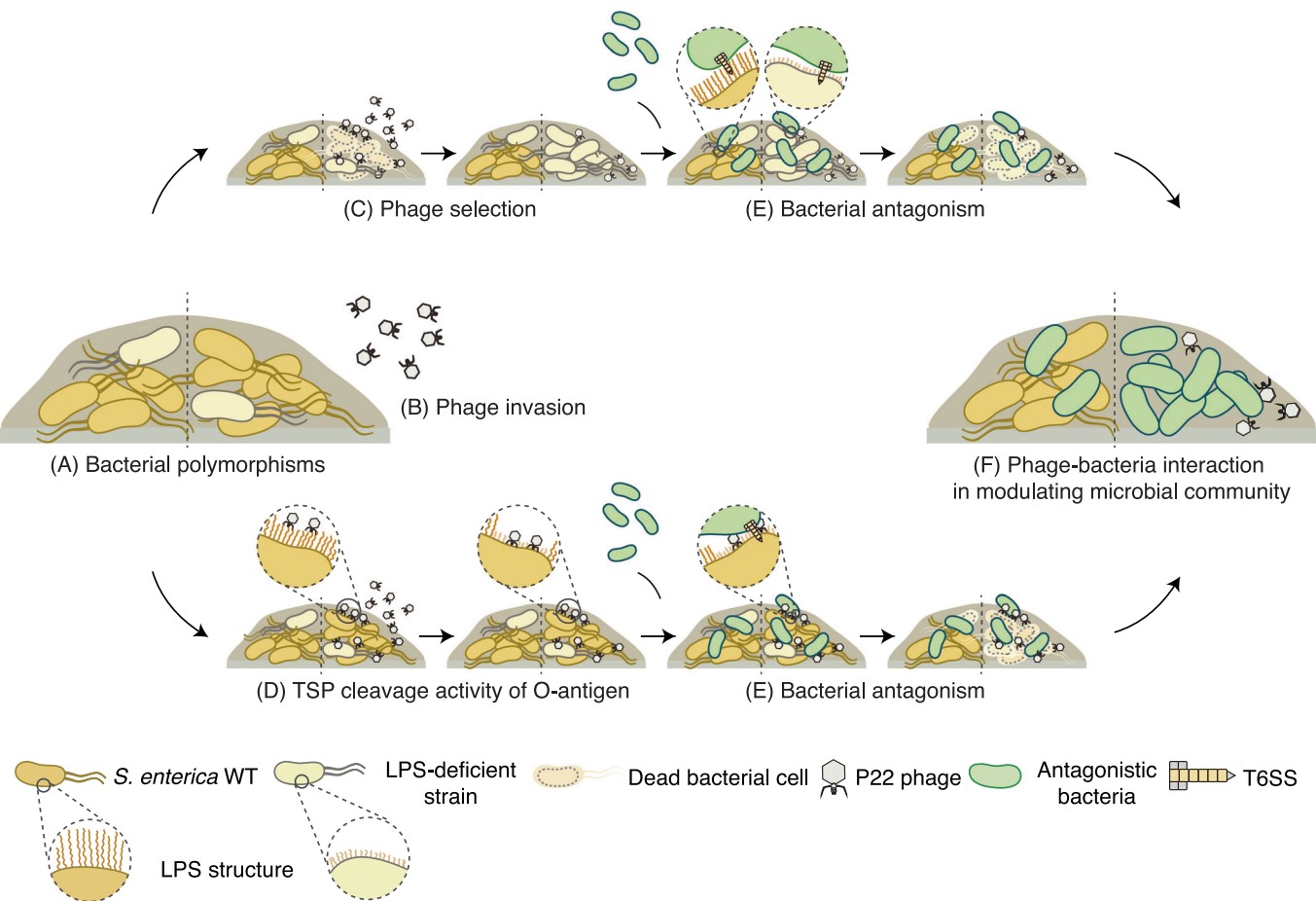

**Figure 7. Working model of how phages modulate interbacterial competition within microbial communities.**

Bacterial interactions are shaped either by the absence (left) or presence (right) of phages. **(A)** Bacteria inhabit polymorphic environments. **(B)** In the presence of P22 phage, bacterial surfaces can be modified via two distinct mechanisms. **(C)** LPS-modified strains are selected due to their resistance to phage adsorption, leading to their dominance within *S. enterica* populations. **(D)** TSP from P22 phage specifically cleaves the bacterial O-antigen of LPS, resulting in a shortened structure. **(E)** LPS deficiency increases the susceptibility of *S. enterica* to contact-dependent killing by bacterial competitors in close proximity. **(F)** Phage-mediated surface modifications influence the outcome of interbacterial competition, driving dynamic shifts in mixed bacterial populations.

deploy specialized weaponry, such as the T6SS, to antagonize rivals and promote niche partitioning. However, in the presence of phages (Fig. 7B), two distinct scenarios could alter this dynamic: (1) bacteria may develop protective modifications on their surfaces to prevent phage adsorption, reducing the likelihood of infection (Fig. 7C); or (2) phages equipped with LPS-targeting endoglycosidases in their TSPs cleave the LPS of prey cells (Fig. 7D). Both scenarios increase the susceptibility of target cells to contact-dependent killing (Fig. 7E), facilitating niche invasion and leading to the competitive exclusion of target populations (Fig. 7F). This phage-driven enhancement of bacterial competition introduces a novel layer of complexity to microbial interactions, emphasizing the dual evolutionary pressures imposed by both phage predation and bacterial antagonism. Our findings underscore the critical role phages play in influencing bacterial competition and shaping microbial community structure. The interplay between bacterial defense mechanisms and phage-mediated pressures indicates that, beyond their traditional role as predators, phages may act as indirect facilitators of interbacterial interactions. This model

establishes a framework for future studies to investigate these dynamics across diverse microbial ecosystems and environmental contexts, which is essential for a comprehensive understanding of microbial ecology.

Further research should focus on how these interactions influence the long-term stability and composition of bacterial populations in natural settings. Since LPS-defective bacterial strains face significant disadvantages in natural environments, the prevalence of such phage-resistant strains in nature remains unclear. Interestingly, LPS have been found to be dispensable in some contexts, and LPS-defective strains can provide advantages under certain conditions. For example, *Neisseria meningitidis* strains depleted of LPS due to an early block in lipid A biosynthesis have been shown to remain viable (Steeghs et al, 2001; Steeghs et al, 1998). These strains may possess adaptive advantages in specific circumstances (Fransen et al, 2009). Similarly, some pathogenic bacteria produce truncated LPS structures lacking O-antigen or introduce modified LPS to evade host immune responses (Maldonado et al, 2016). Evolutionary pressures, such as phage

predation, often drive mutations in O-antigen biosynthesis pathways, resulting in alteration to or an absence of O-antigen (Yu et al, 2024a). Moreover, recent studies have indicated that trade-offs between abiotic and biotic stresses can influence LPS integrity. For instance, LPS-deficient strains in extreme environments may exhibit selective advantages (Burmeister et al, 2020; Carretero-Ledesma et al, 2018). These findings underscore the context-dependent functionality of LPS and their potential dispensability in certain ecological niches.

In addition, the therapeutic potential of phage-derived enzymes, such as TSP, should be explored in greater detail, particularly given the growing global threat of antibiotic resistance. Phage-derived enzymes could offer a novel approach to antimicrobial therapy by targeting bacterial defenses and enhancing the efficacy of conventional treatment strategies (Knecht et al, 2019).

# Methods

### Reagents and tools table

| Reagent/resource | Reference or source | Identifier or catalog number |
|---|---|---|
| **Experimental models** | | |
| Bacterial strains | This study | Table EV1 |
| Bacteriophage strains | This study | Table EV1 |
| **Recombinant DNA** | | |
| pRE118_Sent$\Delta$waaL | This paper | N/A |
| pRE118_Sent$\Delta$waaJ | This paper | N/A |
| pRE118_Sent$\Delta$waaO | This paper | N/A |
| pRE118_Sent$\Delta$waaG | This paper | N/A |
| pRE118_Sent$\Delta$waaK | This paper | N/A |
| pRE118_Sent$\Delta$waaB | This paper | N/A |
| pRE118_Sent$\Delta$waaQ | This paper | N/A |
| pRE118_Sent$\Delta$waaY | This paper | N/A |
| pRE118_Sent$\Delta$wzzB | This paper | N/A |
| pRE118_Sent$\Delta$fepE | This paper | N/A |
| pPSV39_SentWzzB | This paper | N/A |
| pPSV39_SentFepE | This paper | N/A |
| pRE118_Ecoli$\Delta$waaO | This paper | N/A |
| pRE118_Crod$\Delta$waaO | This paper | N/A |
| pETDuet-1-His6_P22TSP | This paper | N/A |
| pETDuet-1-His6_P22TSP D392N | This paper | N/A |
| pGRG37 | McKenzie et al, 2006 | N/A |
| **Oligonucleotides and other sequence-based reagents** | | |
| PCR primers | Mission Biotech | Table EV2 |
| **Chemicals, enzymes, and other reagents** | | |
| Chloramphenicol | Sigma-Aldrich | 56-75-7 |
| Kanamycin | Sigma-Aldrich | 25389-94-0 |
| Streptomycin | Sigma-Aldrich | 3810-74-0 |
| Nalidixic acid | Sigma-Aldrich | 389-08-2 |

| Reagent/resource | Reference or source | Identifier or catalog number |
|---|---|---|
| Gentamycin | Sigma-Aldrich | 1405-41-0 |
| Difco LB broth | BD Biosciences | 244610 |
| BHI medium | Sigma-Aldrich | 75917 |
| Sucrose | Sigma-Aldrich | 57-70-1 |
| Isopropyl-$\beta$-D-thiogalactopyranoside (IPTG) | Sigma-Aldrich | I1284-5ML |
| Gibson Assembly® Master Mix | New England Biolabs | E2611L |
| *XbaI* restriction enzyme | New England Biolabs | R0145L |
| *SacI* restriction enzyme | New England Biolabs | R3156S |
| *BamHI* restriction enzyme | New England Biolabs | R3136S |
| *NotI* restriction enzyme | New England Biolabs | R3189S |
| EZ-Tn5<KAN-2>Tnp Transposome Kit | Epicentre BioTechnologies | TSM99K2 |
| DNase I | Thermo Scientific | EN0521 |
| Mitomycin C | Sigma-Aldrich | 50-07-7 |
| Tris-base | Sigma-Aldrich | PT-CHTS-1KG |
| Sodium dodecyl sulfate (SDS) | Sigma-Aldrich | 151-21-3 |
| $\beta$-mercaptoethanol | Sigma-Aldrich | 60-24-2 |
| Bromophenol Blue | Sigma-Aldrich | 115-39-9 |
| Proteinase K | Invitrogen | 25530049 |
| Uranyl acetate | Sigma-Aldrich | 20816-12-0 |
| Imidazole | Sigma-Aldrich | 288-32-4 |
| Dithiothreitol (DTT) | Roche | 3483-12-3 |
| Protease inhibitor cocktail | Roche | 04693132001 |
| 1-N-phenylnaphthylamine (NPN) | Sigma-Aldrich | 90-30-2 |
| Vancomycin (VAN) | Sigma-Aldrich | 1404-93-9 |
| HEPES | Sigma-Aldrich | 7365-45-9 |
| EDTA | Sigma-Aldrich | 10378-23-1 |
| **Software** | | |
| Fiji | ImageJ | |
| Nanoscope Analysis software | Bruker AXS | |
| Geneious Prime | Dotmatics | |
| GraphPad Prism | Dotmatics | |
| Adobe Illustrator | Adobe | |
| **Other** | | |
| Nitrocellulose filter | Cytiva | |
| Pro-Q Emerald 300 LPS extraction Gel Stain kit | Invitrogen | |
| DNeasy Blood & Tissue Kit | Qiagen | |
| QIAprep Spin Miniprep Kit | Qiagen | |
| Minisart 0.2 μm filters | Sartorius | |
| Minisart 0.45 μm filters | Sartorius | |
| SPRIselect Bead-Based Reagent | Beckman Coulter | |

| Reagent/resource | Reference or source | Identifier or catalog number |
|---|---|---|
| Terminal deoxynucleotidyl transferase (TdT) | New England Biolabs | |
| Infinite F200 series plate reader | Tecan | |
| SpectraMax iD5 microplate reader | Molecular Devices | |
| Carbon-coated formvar film | Ted Pella Inc. | |
| Talos 120 C transmission electron microscope | Thermo | |
| Stratalinker UV Crosslinker 1800 | Stratagene | |
| Ni-NTA agarose | Qiagen | |
| Multimode NanoScope V system | Bruker AXS | |
| Silicon cantilever | FMAuD, NanoSensor | |

## Bacteria, phages, and culture conditions

The bacterial and phage strains used in this study are listed in Table EV1. The following *Escherichia coli* strains were used: DH5α and PIR1 for cloning and plasmid maintenance; S-17-1 λ pir for conjugal transfer of the pRE118 plasmid into *S. enterica*, *E. coli*, and *C. rodentium*; and BL21 (DE3) for overproduction and purification of TSP protein samples. *Salmonella enterica* serovar Typhimurium ATCC-14028, *Enterobacter cloacae* ATCC-13047, enterohemorrhagic *Escherichia coli* EDL933, *Citrobacter rodentium* DBS100, and *Burkholderia thailandensis* E264 were used for bacterial co-culture and competition assays. Bacterial strains were routinely grown in Luria Bertani (LB) medium or Brain Heart Infusion (BHI) medium with appropriate antibiotics at 37 °C and shaking at 200 rpm. Antibiotics were used at the following concentrations: 50 μg/mL for kanamycin, 25 μg/mL for chloramphenicol, 50 μg/mL for streptomycin, 50 μg/mL for nalidixic acid, and 15 μg/mL for gentamycin.

## Plasmid construction

All plasmids used in this study are described in the Reagent and Tools Table. All primers used in our study are listed in Table EV2. In-frame chromosomal deletions in *S. enterica*, *E. coli*, and *C. rodentium* were generated using the pRE118 vector (Edwards et al, 1998). To generate chromosomal mutation constructs, 800 bp regions flanking the mutation site were amplified by polymerase chain reaction (PCR), sewn together, and inserted into the vector using Gibson assembly at *XbaI* and *SacI* restriction sites (Gibson et al, 2009). For the expression and purification of P22 TSP, the region encoding amino acid residues 109–666 was PCR-amplified and inserted into pETDuet-1 with an N-terminal His⁶-tag using Gibson assembly at the *BamHI* and *NotI* restriction sites. The TSP catalytic mutant D392N was generated via site-directed mutagenesis (Kunkel, 1985). To overexpress *wzzB* and *fepE*, the respective genes were amplified by PCR and inserted into the inducible pPSV39 vector via Gibson assembly at the *XbaI* and *SacI* restriction sites (Lee et al, 2010).

## Generation of mutant strains of *S. enterica*, *E. coli*, and *C. rodentium*

To generate deletions in *S. enterica*, *E. coli*, and *C. rodentium*, mutation constructs cloned into pRE118 were transformed into conjugative

*E. coli* S-17 λ pir (Edwards et al, 1998). Conjugative donor cells carrying the deletion construct and recipient cells were grown overnight on LB plates containing the appropriate antibiotics. The donor and recipient cells were then mixed at a 10:1 ratio, spotted onto LB agar plates, and incubated at 37 °C for 6 h to allow transfer of the deletion plasmid via conjugation. The cell mixtures were scraped into LB medium and plated onto LB agar supplemented with kanamycin and chloramphenicol (for *S. enterica* and *E. coli*) or kanamycin and nalidixic acid (for *C. rodentium*) to select for cells containing the chromosomally inserted deletion construct. The merodiploid strains were grown overnight in non-selective LB medium at 37 °C, followed by counter-selection on nutrient agar supplemented with 10% (w/v) sucrose at room temperature. Kanamycin-sensitive colonies were picked, and the deletions were confirmed by colony PCR and Sanger sequencing. The chloramphenicol-resistant *S. enterica* and *E. coli* strains were generated using pGRG37 methods (McKenzie and Craig, 2006).

## Phage propagation

*S. enterica* ATCC-14028 was used as the host for phage propagation to obtain a high-titer phage lysate (>10⁸ PFU). An overnight culture was diluted in 50 mL of fresh LB broth to an OD₆₀₀ of 0.015 and incubated at 37 °C until the OD₆₀₀ reached 0.2. Phage lysate was then added at a low multiplicity of infection (MOI = 0.1), and the culture was incubated at 37 °C for 6 h. After incubation, the mixture was centrifuged at $6000 \times g$ for 10 min and filtered through a 0.2-μm filter. The phage lysate was stored at 4 °C until use, and the phage titer was determined by spot assays.

## Isolation of bacteriophage from soil

Soil samples were collected from Nangang District in Taipei City. Each soil sample was suspended in 50 mL of phage resuspension buffer (10 mM Tris-HCl, pH 7.5, 10 mM MgSO₄, 68.5 mM NaCl, 1 mM CaCl₂) and maintained on a rotator overnight to release the phage particles into the buffer. The samples were centrifuged at $7000 \times g$ for 15 min at 4 °C, and the supernatants were collected through 0.45-μm syringe filters.

For phage enrichment and single phage purification, 500 μL of host bacteria growing in the early stationary phase was added to the samples along with fresh BHI medium. The samples were supplemented with 10 mM CaCl₂ to enhance phage growth and incubated overnight at 37 °C, shaking at 120 rpm. The enriched samples were then centrifuged at $13,000 \times g$ for 15 min at 4 °C, and the supernatants were filtered through 0.45-μm syringe filters to obtain the phage suspension. The double-layer agar method was then used to obtain single phage plaques. A single plaque was selected and stored in BHI broth supplemented with 20% (v/v) glycerol at −80 °C.

## Phage genome extraction and sequencing

In all, 20 mL of high-titer phage lysate was treated with DNase I (Thermo, Waltham, USA) and RNase (Thermo, Waltham, US) at a final concentration of 10 μg/mL to remove host DNA at 37 °C for 1 h and then precipitated using PEG/NaCl overnight at 4 °C. The phage lysate was centrifuged at $10,000 \times g$ at 4 °C for 10 min and precipitated phage lysate was then resuspended in buffer ATL, followed by gDNA extraction using a DNeasy Blood & Tissue Kit

(Qiagen). Phage genomic DNA was quantified using a Qubit 3.0 Fluorometer and a Qubit dsDNA BR Assay Kit (Thermo Fisher Scientific, Carlsbad, CA, USA). Subsequently, libraries were prepared for sequencing using the Nanopore Native Barcoding Kit 24 V14 (SQK-NBD114.24). Sequencing was performed using a Nanopore FLO-MIN114 R10 flow cell.

## Spot assay

To calculate the phage titer, 100 μL of mid-log phase *S. enterica* ($OD_{600} = 0.5 – 0.8$) was mixed with either 4 mL of prewarmed 0.7% (*w/v*) soft LB agar or 8 mL of 0.5% (*w/v*) soft BHI agar and immediately poured onto 1.5% (*w/v*) regular LB agar plates. Then, 2 μL of tenfold serially diluted phage lysate was spotted onto the bacterial lawn. The plates were allowed to dry at room temperature and subsequently incubated at 37 °C overnight. The number of visible single plaques in each spot was counted to calculate the phage titer.

## Transposon mutant library construction

The *S. enterica* transposon mutant library was constructed as described previously, with slight modifications (Mandal et al, 2021). In brief, *S. enterica* cultures were harvested at mid-log phase ($OD_{600} = 0.5$) and gently washed with sterilized 10% (*w/v*) glycerol. The EZ-Tn5<KAN-2> Tnp transposome complex (Epicentre BioTechnologies) was added to the electrocompetent cells, which were incubated on ice for 10 min. The mixture was then transferred to an ice-cold 1-mm electroporation cuvette and electroporated at 2450 V. Prewarmed SOC medium was added, and the cells were allowed to recover at 37 °C for 90 min. Tn5 mutants were selected on LB agar plates containing kanamycin, pooled, and stored in LB medium with 50% (*w/v*) glycerol at −80 °C. Approximately 12,000 Tn5::KanR insertion mutants were collected.

## Isolation of phage-resistant variants

The *S. enterica* transposon library was inoculated into fresh LB broth containing kanamycin and incubated at 37 °C until the culture reached the early log phase ($OD_{600} = 0.2$). Phage lysate containing $10^8$ PFU was then added to the culture, which was incubated at 37 °C for 1 h. The $OD_{600}$ was monitored hourly until the culture reached mid-log phase ($OD_{600} = 0.5$). Bacterial samples from the infected culture were plated on LB agar containing kanamycin and incubated at 37 °C overnight. Colonies of *S. enterica* that grew on the plates were picked and tested for phage resistance by spot assay. To confirm that the phage-resistant variants were not lysogens, the putative phage-resistant cells were treated with mitomycin C and plated on an *S. enterica* lawn for further analysis.

## Identification of phage-resistant mutations

The workflow for identifying phage-resistant mutations is depicted in Fig. EV2A. Phage-resistant strains were grown in LB broth at 37 °C overnight, followed by genomic DNA (gDNA) extraction using a DNeasy Blood & Tissue Kit (Qiagen). Primer extension with single primer (EzTn_s) was performed to amplify a fragment of the Tn5-gDNA junction. The extension cycle included a manual hot start, with initial denaturation at 95 °C for 2 min, followed by

the addition of polymerase. The linear extension products were purified using SPRIselect Bead-Based Reagent (Beckman Coulter). A deoxycytidine homopolymer tail (C-tail) was added to the 3' end of the linear extension products using terminal deoxynucleotidyl transferase (TdT; New England Biolabs). The reaction was carried out at 37 °C for 1 h, followed by enzyme inactivation at 75 °C for 20 min. The C-tailed products were then purified again using SPRIselect Bead-Based Reagent. The purified C-tailed products were enriched by PCR using Tn5_F and Tn5_R primers. Finally, the PCR products were heated at 65 °C for 15 min and analyzed on 1.5% agarose gels. DNA fragments ranging from 300 to 500 bp were gel extracted for Sanger sequencing and BLAST analysis.

## Monoculture bacterial growth in LB broth

Bacteria were cultured overnight and subsequently diluted to an $OD_{600}$ of 0.1 in LB broth. A total of 100 μL of the diluted cells was transferred to 96-well, clear-bottom microplates, and the growth rate was measured every 10 min for 12 h using an Infinite F200 series plate reader (Tecan). To prevent cell aggregation, 10 s of shaking was applied prior to each measurement. Blank media and the wild-type strain were included on each plate, and at least three biological replicates were measured for each sample.

## Monoculture bacterial growth on solid agar

Overnight cultures grown in LB medium were pelleted, washed, and adjusted to an $OD_{600} = 20$. Cell mixture (5 μL) was spotted onto nitrocellulose filters placed on LB agar plates and incubated at 37 °C for 12 h. Cells were then scraped from the nitrocellulose and resuspended in 200 μL of LB broth. Initial and post-co-culture cell counts were determined by performing tenfold serial dilutions and plating the dilutions on LB agar plates containing the appropriate antibiotic.

## LPS analysis

LPS was analyzed as previously described, with some modifications (Liu et al, 2016). In brief, overnight bacterial culture ($OD_{600} = 10$) was pelleted and resuspended in 150 μL of Dissociation Buffer A (0.5 M Tris-HCl pH 6.8, 10% (*w/v*) glycerol, 10% (*w/v*) SDS, 5% (*v/v*) β-mercaptoethanol), followed by boiling for 10 min. After cooling to room temperature, the samples were centrifuged at $18,000 \times g$ for 15 min. The supernatant (100 μL) was diluted into 900 μL of Dissociation Buffer B (0.5 M Tris-HCl pH 6.8, 10% (*w/v*) glycerol, 0.05% (w/v) bromophenol blue) supplemented with 1 μL of 20 mg/mL proteinase K, and incubated at 37 °C for 1 h. The samples were separated on 12% or 20% (*w/v*) SDS-PAGE and stained using a Pro-Q Emerald 300 Lipopolysaccharide Gel Stain Kit (Invitrogen) for imaging.

## Transmission electron microscopy (TEM) analysis

For TEM analysis, 4 μL droplets of phage lysate (~$10^{10}$ PFU/mL) were placed on copper grids (300 mesh) coated with a carbon-coated formvar film (Ted Pella Inc., Redding, CA). The phage lysate was allowed to settle for 1 min at room temperature and then dried using 3-M filter paper. Negative staining was performed with 1% (*w/v*) uranyl acetate for 30 s, after which the excess stain was

removed with 3 M filter paper. The sample was gently washed three times with sterilized distilled deionized water, and the grid was air-dried overnight at room temperature. Phage morphology was visualized using a Talos 120 C transmission electron microscope (Thermo, Waltham, USA).

## Bacterial co-culture experiments

Overnight cultures grown in LB medium were pelleted, washed, and adjusted to an $OD_{600}$ of 5 for liquid co-culture assays or to an $OD_{600}$ of 20 for assays on solid agar. Cells were mixed at a 1:1 ratio in LB broth and incubated at 37 °C for 1 h. For solid agar experiments, 5 µL of the cell mixture was spotted onto nitrocellulose filters placed on LB agar plates and incubated at 37 °C for 3 h. Cells were then scraped from the nitrocellulose and resuspended in 200 µL of LB broth. Initial and post-co-culture cell counts were determined by performing tenfold serial dilutions and plating the dilutions on LB agar plates containing the appropriate antibiotics. Relative survival of the bacteria was calculated as the ratio of post-co-culture colony forming units (CFUs) to the average CFUs of wild-type S. enterica.

## Macroscopic cell aggregation assay

Macroscopic cell aggregation assays were performed as described previously, with slight modifications (Chen et al, 2024). Bacterial cultures were grown overnight at 37 °C in LB broth containing kanamycin. After incubation, the cultures were vortexed and transferred to a 2-mm cuvette. The bacterial cultures were then allowed to settle at room temperature for 22 h to observe cell aggregation.

## Bacterial competition assays

Competition assays were performed as described previously, with slight modifications (Ting et al, 2020). The bacterial strains used in the assays were cultured overnight at 37 °C, harvested by centrifugation, washed, and resuspended in LB medium. E. cloacae or B. thailandensis were used as the attacker strains, whereas S. enterica, E. coli, and C. rodentium served as the prey strains. Attacker and prey cells were mixed at a 10:1 ratio (200:20 $OD_{600}$), and a 5 µL aliquot of the mixture was spotted onto nitrocellulose filters placed on prewarmed LB agar plates and incubated at 37 °C for 1 h. For assays involving wzzB or fepE overexpression strains, the mixtures were spotted on LB agar plates supplemented with 0.5 mM IPTG. Following incubation, cells were scraped from the nitrocellulose filters, resuspended in 200 µL of LB broth, and serially diluted (tenfold) for CFU counting on LB agar plates with appropriate antibiotics. The bacterial relative protective index was calculated as the ratio of the output prey-to-attacker ratio divided by the input prey-to-attacker ratio, followed by normalization to the protective index of wild-type S. enterica.

To investigate the effect of phage-derived LPS modifiers on bacterial susceptibility to T6SS-mediated attacks, wild-type S. enterica cells were treated with purified TSP (10 nM), live phages (MOI = 100), or UV-inactivated phages (MOI = 100) for 10 min at 37 °C. Treated cells were then mixed with E. cloacae at a 10:1 attacker-to-prey ratio, and the mixture was spotted onto nitrocellulose filters placed on prewarmed LB agar plates. After

incubation at 37 °C for 1 h, cells were collected by scraping them from the filters and resuspending them in 200 µL of LB broth. Initial and post-co-culture CFU counts were determined through serial dilutions and plating on LB agar with selective antibiotics. The bacterial relative protective index was calculated as the ratio of the output prey-to-attacker ratio divided by the input prey-to-attacker ratio and normalized to the protective index of wild-type S. enterica.

## P22 TSP purification

Purification of recombinant P22 TSP (wild-type and D392N mutant) was performed as described previously (Miller et al, 1998). In brief, the N-terminal His[6]-tagged TSP was expressed in E. coli BL21 (DE3). An overnight culture was diluted 100-fold into 1 L of LB medium containing carbenicillin and incubated at 37 °C until the culture reached mid-log phase ($OD_{600} = 0.6 - 0.8$). TSP expression was induced by adding 1 mM IPTG and incubating at 37 °C for 4 h. Cells were harvested by centrifugation at 4 °C and resuspended in 20 mL of resuspension buffer (20 mM Tris-HCl pH 7.5, 300 mM NaCl, 5 mM imidazole pH 7.0, 1 mM DTT, and a protease inhibitor cocktail). The resuspended cells were disrupted by sonication, and cell debris was removed by centrifugation at $40,000 \times g$ for 30 min at 4 °C. The soluble His[6]-TSP was purified from the supernatant using gravity flow through a 2 mL Ni-NTA agarose column. Bound proteins were eluted using a linear imidazole gradient (25–300 mM). Purified proteins were analyzed by 12.5% (w/v) SDS-PAGE followed by Coomassie Brilliant Blue staining.

## Phage inactivation by UV treatment

The purified phage stock was diluted to ~$10^9$ PFU/mL, and 1 mL aliquots were placed in Petri dishes covered with parafilm. The dishes were exposed to 254 nm UV light using a Stratalinker UV Crosslinker 1800 (Stratagene, La Jolla, CA) for the indicated time intervals. Phage infection efficiency following UV treatment was assessed by spot assay (Fig. 5E; Appendix Fig. S2C,D).

## Membrane permeability assays

Outer membrane permeability was assessed using NPN uptake and VAN sensitivity assays, as described previously (Helander and Mattila-Sandholm, 2000; Muheim et al, 2017). For the NPN uptake assay, mid-log phase cultures ($OD_{600} = 0.5$) were pelleted and gently washed with 5 mM HEPES buffer (pH 7.2). An equal volume of HEPES buffer containing 20 µM NPN was added to the samples and mixed thoroughly. As a control, 1 mM EDTA was added to wild-type cells. Fluorescence was measured using an optical-bottom black plate in a SpectraMax iD5 microplate reader (Molecular Devices) with excitation at 350 nm and emission at 420 nm, recorded at 30-s intervals for 10 min. Data collected over 10 min were averaged, and NPN uptake activity was normalized by subtracting the background fluorescence in the absence of NPN. For the VAN sensitivity assay, overnight cultures were diluted to an $OD_{600}$ of 0.05 in 25 mL of LB broth, followed by the addition of 150 µg/mL vancomycin. As a control, 1 mM EDTA was added to wild-type cells. After 2 h of incubation at 37 °C, CFUs were determined by performing tenfold serial dilutions and plating on LB agar.

## Atomic force microscopy (AFM)

Atomic force microscopy sample preparation was performed as described previously, with some modifications (Oh, 2022). An overnight bacterial culture was diluted 100-fold and incubated at 37 °C until it reached mid-log phase ($OD_{600} = 0.6 - 0.8$). Cells were harvested by centrifugation at $OD_{600} = 1$, washed, and then resuspended in 10 mM HEPES buffer (pH 7.4) (Singh et al, 2022). For immobilization, 200 μL of the bacterial suspension was applied to freshly cleaved mica and incubated for 20 min at room temperature. Following immobilization, the sample was gently rinsed with Milli-Q water to remove any non-adherent cells, then air-dried in a desiccator for 30 min prior to AFM measurements.

AFM measurements were carried out using a Multimode NanoScope V system (Bruker AXS) with PeakForce Tapping (PFT) and Force Volume (FV) mapping modes. A commercial tip holder equipped with silicon (Si) cantilevers (FMAuD, NanoSensor), a spring constant of 2.0 N/m, and a resonance frequency of 70.0 kHz was employed. PFT mode was used for high-resolution imaging with a peak force setpoint of 1 nN and modulation of 2 kHz in the Z-ramp. For mechanical mapping and force spectroscopy, FV mode was employed to collect force-distance (F-Z) curves across the sample surface. Force spectroscopy was conducted using $32 \times 32$ pixels in the XY scan area with 512 points per force curve at a 3 Hz Z-ramp for data acquisition and analysis. The force-distance curves were calibrated from the original force-displacement (F–D) data following standard processing protocols (Gavara, 2017). AFM imaging was restricted to isolated bacterial cells to prevent confounding effects from cell aggregation. The AFM instrument was housed in a humidity-controlled environment (~35%) to reduce capillary forces between the AFM tip and the bacterial surface. Raw AFM images were processed and analyzed using Nanoscope Analysis software (Bruker AXS).

To analyze force-distance curves, the initial jump-in force (point 1) was defined as the contact point on the LPS layer, corresponding to a tip-sample distance of 0 nm and a tip-sample interaction force of 0 nN. Two distinct interaction profiles from each force curve were identified by slope changes, in which the increase of the slope represented contact with a stiffer substrate. The length of the LPS layer was measured as the distance between point 1 and point 2 (Fig. 6G). Ten force curves were obtained from multiple cells at different locations to ensure representative sampling.

### Quantification and statistical analysis

Statistical significance in bacterial co-culture and competition assays was assessed by unpaired two-tailed *t* tests between relevant samples. Statistical significance are provided in the figure legends.

## Data availability

Phage genome data associated with this study is available in the NCBI BioProject resource under accession number PRJNA1199570.

The source data of this paper are collected in the following database record: biostudies:S-SCDT-10_1038-S44318-025-00406-3.

## Peer review information

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

## Acknowledgements

The authors thank Dr. Waldemar Vollmer, Dr. Marcin Grabowicz, Dr. Harmit Malik, Dr. Meng-Chiao Ho, and Ting laboratory members for helpful discussions; Dr. Joseph Mougous, Dr. Shu-Jung Chang, Dr. Hsin-Hung Chou, Dr. Hung-Ta Chen, and Dr. Sue Lin-Chao for sharing reagents; Dr. Soo-Chen Cheng and Dr. Jun-Yi Leu for sharing equipment; Dr. Shu-Yun Tung, Dr. Hsin-Nan Lin, Dr. Chen-Hsin Yu, Ms. Wen-Li Pon, and Ms. Shin-Yi Du for technical assistance; and Dr. John O'Brien for manuscript editing. The authors acknowledge funding support from Academia Sinica (AS-CDA-112-L05 to S-YT) and the National Science and Technology Council, Taiwan (111-2311-B-001-006-MY2, 112-2311-B-001-020, and 113-2628-B-001-008-MY3 to S-YT).

## Author contributions

**Chia-En Tsai**: Conceptualization; Data curation; Formal analysis; Methodology; Writing—original draft; Writing—review and editing. **Feng-Qi Wang**: Conceptualization; Data curation; Formal analysis; Methodology; Writing—original draft; Writing—review and editing. **Chih-Wen Yang**: Conceptualization; Data curation; Formal analysis; Methodology; Writing—original draft. **Ling-Li Yang**: Data curation. **Thao VP Nguyen**: Data curation. **Yung-Chih Chen**: Data curation. **Po-Yin Chen**: Writing—original draft; Writing—review and editing. **Ing-Shouh Hwang**: Conceptualization; Methodology. **See-Yeun Ting**: Conceptualization; Data curation; Formal analysis; Supervision; Funding acquisition; Writing—original draft; Writing—review and editing.

Source data underlying figure panels in this paper may have individual authorship assigned. Where available, figure panel/source data authorship is listed in the following database record: biostudies:S-SCDT-10_1038-S44318-025-00406-3.

## Disclosure and competing interests statement

The authors declare no competing interests.

# Expanded View Figures

**Figure EV1. Validation of phage resistance in isolated mutants and monocultural growth experiments.**

(**A**) TEM images of model phages used in the study: Felix O1 (left), P22 (middle), and Chi (right). Scale bar = 0.1 µm. (**B**) Tenfold serial dilution spot assay of phages Felix O1 (left), P22 (middle), and Chi (right) on the indicated phage-resistant *S. enterica* strains. The image is representative of triplicate experiments. (**C**) Growth curves of the indicated phage-resistant *S. enterica* strains in LB broth at 37 °C. Wild-type, Δ*waaL*, and Δ*waaG* strains are shown as controls. (**D**) CFUs of the indicated phage-resistant *S. enterica* strains grown at 37 °C for 12 h on LB agar plates. Wild-type, Δ*waaL* (ΔL), and Δ*waaG* (ΔG) strains are shown as controls.

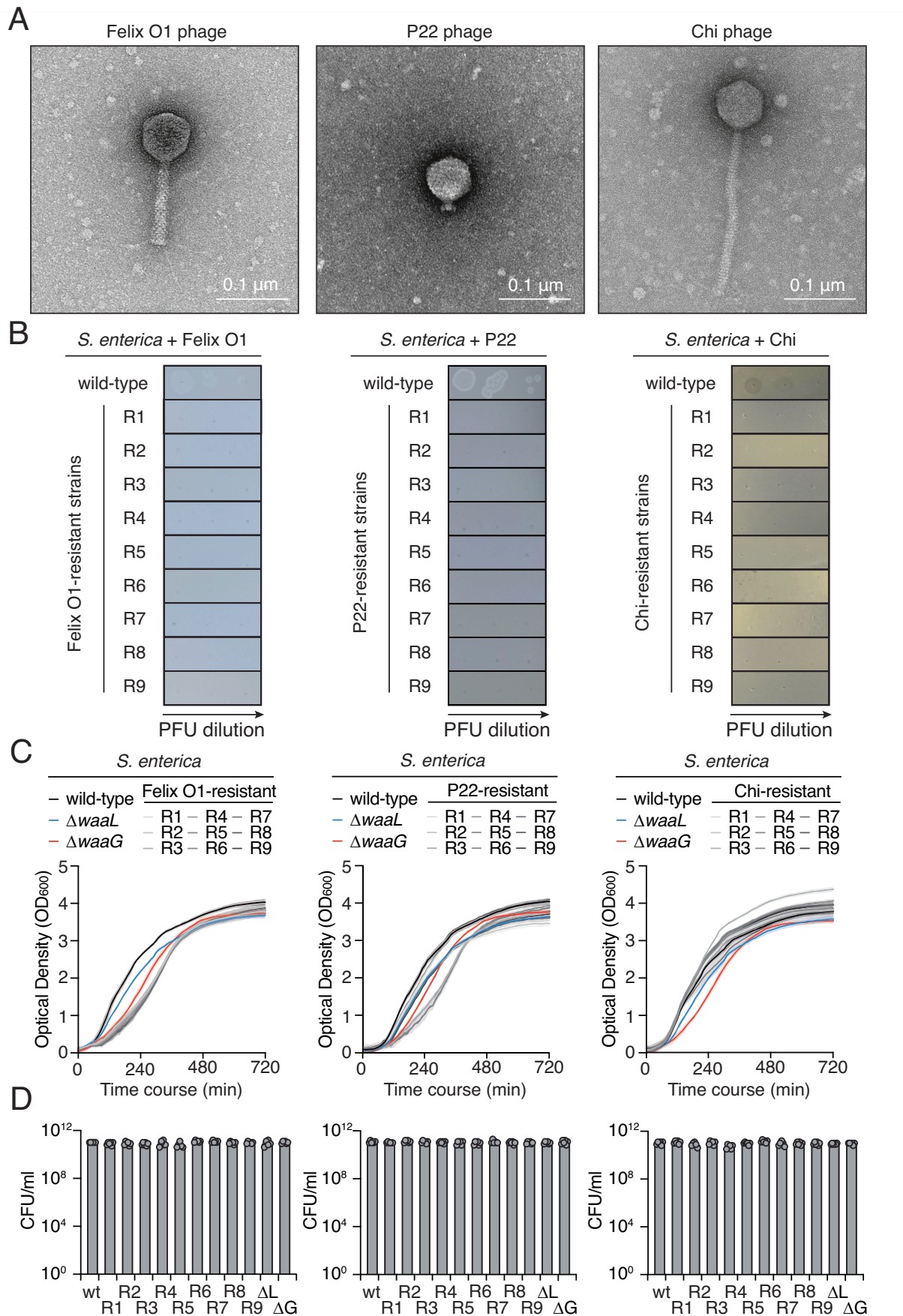

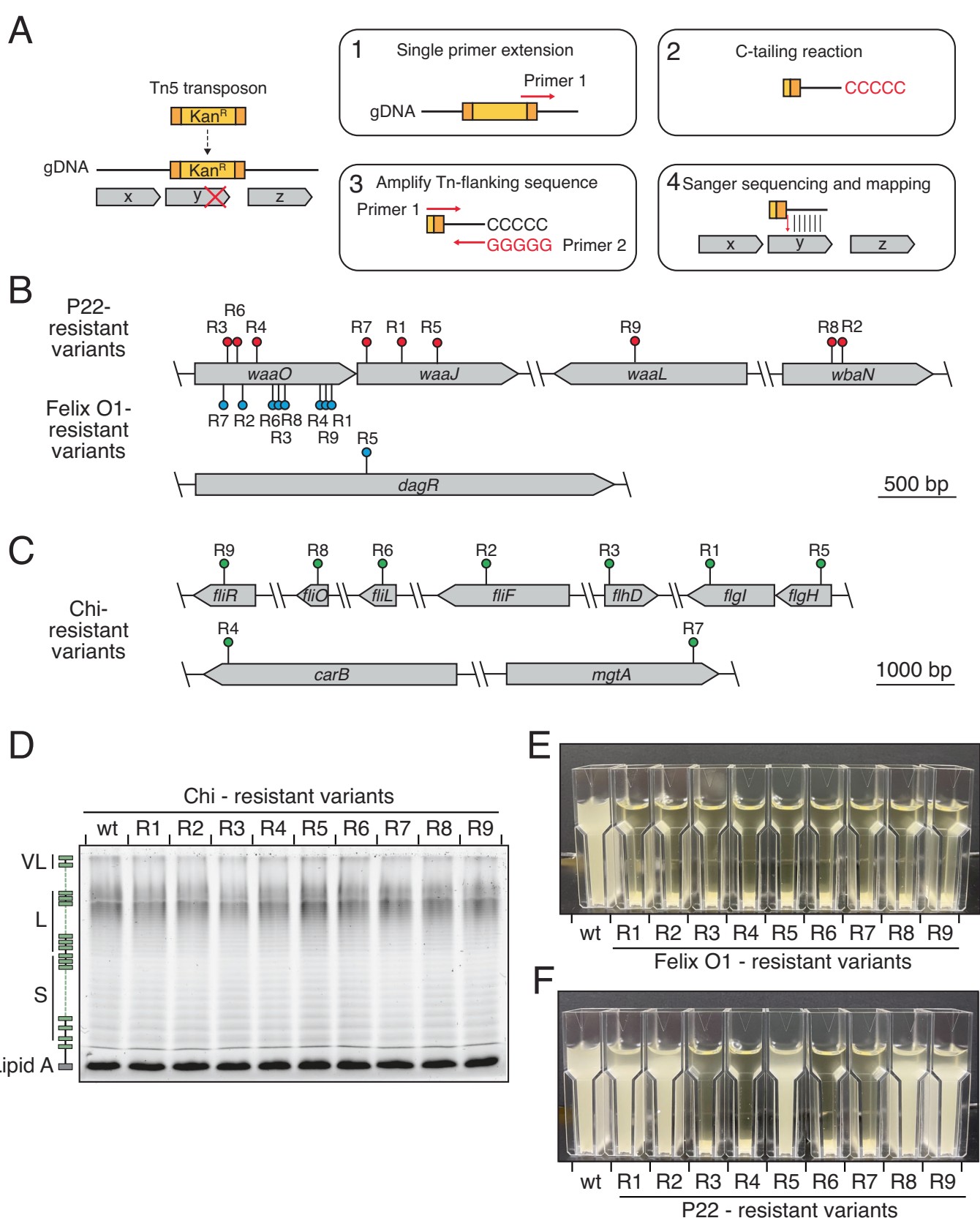

◀ **Figure EV2. Characterization of transposon insertion mutations in phage-resistant *S. enterica* isolates.**

(**A**) Schematic workflow for identifying phage-resistant mutations. Detailed procedures are provided in the Methods section. (**B**) Mutated genes in the P22-resistant and Felix O1-resistant *S. enterica* isolates. Scale bar = 500 bp. (**C**) Mutated genes in the Chi-resistant *S. enterica* isolates. Scale bar = 1000 bp. (**D**) 12% LPS PAGE profiles showing LPS levels from the indicated Chi-resistant *S. enterica* strains. A schematic representation of LPS is shown on the left. (**E, F**) Macroscopic aggregation analysis to assess LPS deficiency in the indicated Felix O1-resistant (**E**) or P22-resistant (**F**) *S. enterica* strains. The images are representative of triplicate experiments.

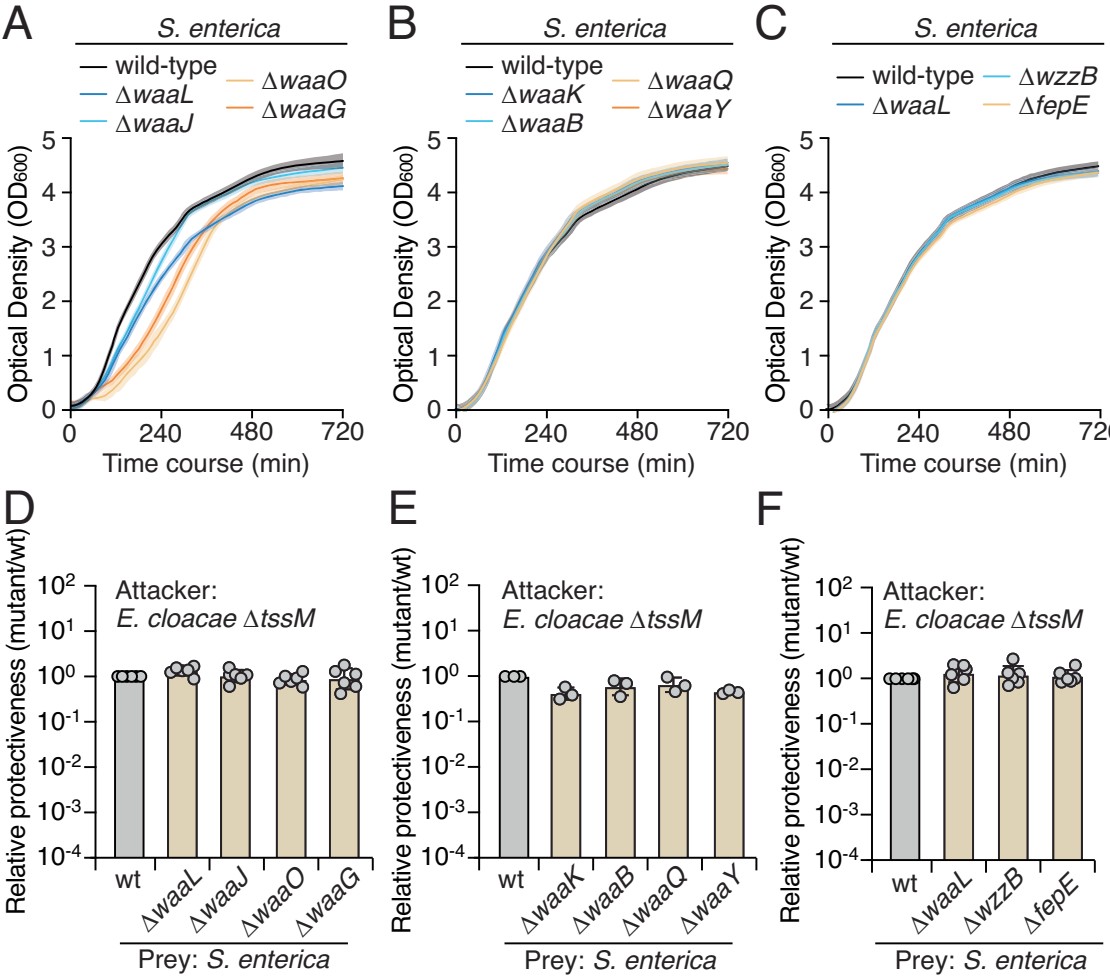

**Figure EV3.  LPS-deficient *S. enterica* mutants exhibit no significant growth or competition defects.**

(**A–C**) Growth curves of the indicated *S. enterica* LPS mutants in LB broth at 37 °C. The wild-type strain is shown as a control. (**D–F**) Protectiveness of the indicated *S. enterica* mutant strains relative to the wild-type strain after competition with *E. cloacae* ΔtssM. Data in (**D–F**) are presented as means ± SD (*n* = 6).

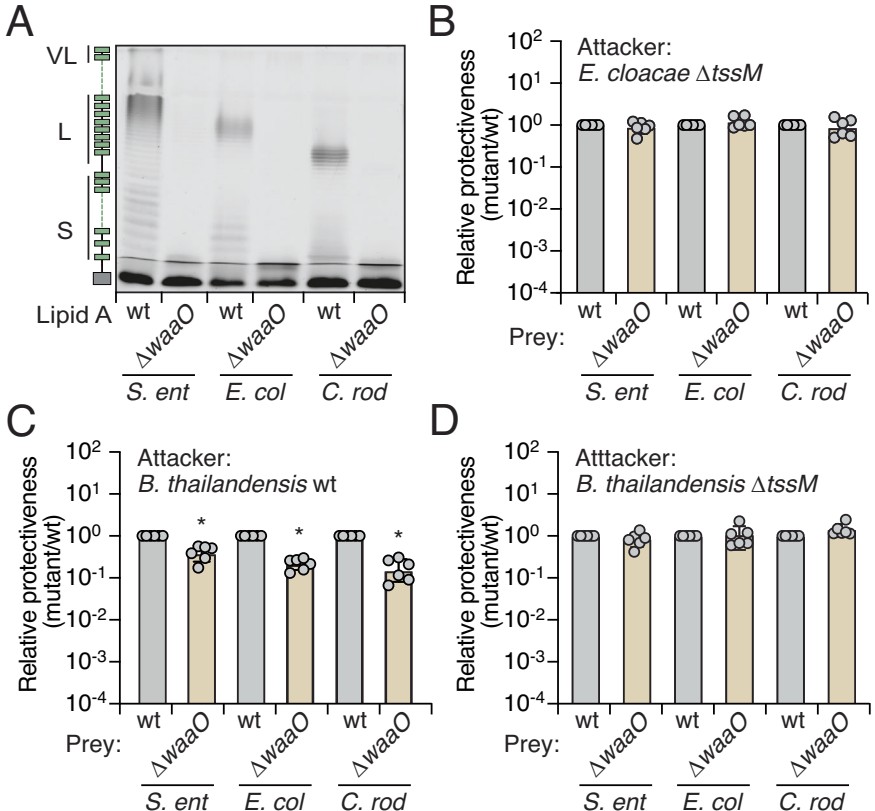

**Figure EV4. LPS-deficient bacteria exhibit fitness deficits in competition with antagonistic bacteria carrying active T6SS.**

(A) 12% LPS PAGE profile displaying LPS levels produced by the indicated *S. enterica*, *E. coli*, and *C. rodentium* strains. (B) Protectiveness of the indicated bacterial species (Δ*waaO* mutant) relative to the wild-type strain after competition with *E. cloacae* Δ*tssM*. (C, D) Protectiveness of the indicated bacterial species (Δ*waaO* mutant) relative to the wild-type strain after competition with *B. thailandensis* wild-type (C) or Δ*tssM* (D) strains. Data in (B–D) are presented as means ± SD (*n* = 6). Asterisks indicate statistically significant differences in the relative protective index between the mutant and wild-type strains (*P* < 0.05).

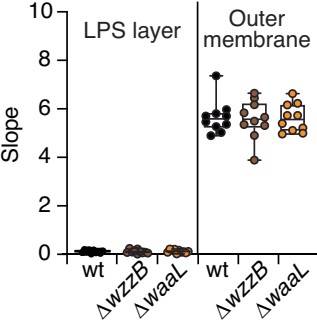

**Figure EV5.  The slope of AFM force-distance curves.**

Two distinct slopes from each force-distance curve were obtained for the *S. enterica* wild-type, Δ*wzzB*, and Δ*waaL* strains. Ten curves were divided into two segments by point 2, as shown in Fig. 6G. The slope measured from the first segment corresponds to the LPS layer (left), whereas the second segment represents the outer membrane (right).

