## [Peer Review File · The EMBO Journal]

Surface-mediated Bacteriophage Defense Incurs Fitness Tradeoffs for Interbacterial Antagonism

Chia-En Tsai, Feng-Qi Wang, Chih-Wen Yang, Ling-Li Yang, Thao Nguyen, Yung-Chih Chen, Po-Yin Chen, Ing-Shouh Hwang, and See-Yeun Ting

Corresponding author(s): See-Yeun Ting (syting@gate.sinica.edu.tw)

Review Timeline:

Transfer from Review Commons:	24th Dec 24
Editorial Decision:	14th Feb 25
Revision Received:	25th Feb 25
Accepted:	27th Feb 25

Review
COMMONS

Editor: Hartmut Vodermaier

Transaction Report:

Review #1

1. Evidence, reproducibility and clarity:

Evidence, reproducibility and clarity (Required)

This manuscript by Tsai et al. shows that phage resistance mutations (LPS truncation) confer a cost during interbacterial competition. The authors show that various phage resistant mutants of *S. enterica* are inhibited by *E. cloacae* in a contact dependent manner (on a solid surface but not in liquid). Further experiments showed that this inhibition of *S. enterica* was mediated by T6SS in *E. cloacae*. The authors then dissect which parts of the LPS are required for resistance against T6SS attacks and show that a similar resistance is conferred against T6SS of *B. thailandensis* and *C. rodentium*. Moreover, the authors show that enzymatic degradation of LPS by a phage enzyme can also increase sensitivity to T6SS (including when such enzymes are on phage particles). Finally, the authors suggest that the change in the thickness of the LPS surface layer could be the reason for changes in T6SS susceptibility. Overall, the manuscript is very well-written. The experiments and controls are explained in sufficient detail and in a logical order. The figures are clear and easy to navigate. The findings are very interesting and important for the T6SS field but also for general understanding how different evolutionary pressures combine and influence each other. I believe that this manuscript will initiate further research in this direction.

The only major point that I would like to raise is that I am not generally convinced that the 2 nm difference in the thickness of LPS is the main reason for the observed differences in T6SS-mediated killing of *S. enterica*. Based on what we know about T6SS mode of action, we expect that it is potentially pushing effectors by up to several hundreds of nanometers. Therefore, the change in the LPS thickness by a few nanometers (as measured by AFM) seems insufficient to provide enough spacing between the attacker and the prey to significantly decrease T6SS effector delivery. While it is clear that understanding the exact reason for the LPS mediated resistance is beyond the scope of this manuscript, I would suggest that the authors consider the fact that T6SS is known to deliver proteins even to the cytoplasm of target gram-negative cells and discuss the mode of action of the machine in the context of their finding. If the T6SS was drawn to scale in the model figure, it would become apparent that 2 nm change in the distance between two cells has probably no major impact on killing by T6SS and the actual reason for the observed phenotype is likely more complicated than what is proposed.

****Minor:****

Specify which T6SS of *B. thailandensis* was tested.

Use a different naming of the two strains used in competition assays than "donor" and "recipient".

Indicate in the material and methods ODs of bacterial mixtures used in the "Bacterial competition assays".

2. Significance:

Significance (Required)

This manuscript is interesting for researchers who study T6SS, phage predation and other evolutionary pressures shaping bacterial interactions. The work provides new and interesting insights. My expertise in LPS biology is limited.

3. How much time do you estimate the authors will need to complete the suggested revisions:

Estimated time to Complete Revisions (Required)

(Decision Recommendation)

Less than 1 month

No

Review #2

1. Evidence, reproducibility and clarity:

Evidence, reproducibility and clarity (Required)

This work investigates the fitness trade-offs in *Salmonella enterica* resistant to phages. The authors performed co-culture experiments with *S. enterica*, *E. coli*, and *E. cloacae* and found that phage-resistant *S. enterica* strains displayed reduced fitness in the presence of *E. cloacae*. Further experiments demonstrated that phage-resistant *S. enterica* strains were more susceptible to the type VI secretion system (T6SS) of *E. cloacae*. The authors then examined the role of the O-antigen of lipopolysaccharide (LPS) in T6SS-mediated interbacterial antagonism. By constructing *S. enterica* mutants with varying O-antigen chain lengths, the authors demonstrated that the O-antigen protects *S. enterica* from T6SS attack. They then demonstrated that the O-antigen-deficient *S. enterica*, *E. coli*, and *C. rodentium* strains were more susceptible to T6SS attack by *E. cloacae*. Finally, the authors showed that phage tail spike proteins (TSPs) with endoglycosidase activity could cleave the bacterial O-antigen, thereby increasing susceptibility to T6SS attack.

The study is well-designed and the experiments are well-executed. The findings are significant and have implications for the understanding of microbial community dynamics.

****Major comments:****

While the study elegantly demonstrates the link between phage resistance, LPS structure, and T6SS susceptibility, we must remember that these LPS-defective strains are likely at a significant

disadvantage in real-world environments without the influence of competing bacteria. Whether it's the gut or external environments, Salmonella needs its LPS for protection against a myriad of host and environmental factors. It seems a bit redundant for T6SS mediated antagonism to select for LPS structures when those structures are essential for bacterial survival outside of this very specific context. It would benefit some discussion about the likelihood of these phage-resistant, LPS-defective strains actually persisting and competing effectively in a more natural setting.

****Minor comments****

Figure 5 could be more effective if panels b and C are together

#69 Authors should define mucoid

#155 Authors should explain that this result is expected since T6SS acts on solid surface while CDI works in liquid cultures

#clarify what it is meant by unicellular cultures. Should it be monocultures?

#618 add to the text how much dead phage was added per bacterial cell

#364 references needed for "consistent with predictions for intact LPS structures "

2. Significance:

Significance (Required)

This study offers a new perspective on the interplay between phage resistance and bacterial fitness in the context of microbial communities. While the concept of fitness trade-offs associated with antibiotic resistance is well-established, the authors extend this paradigm to phage resistance. They demonstrate that phage-resistant *Salmonella enterica* strains exhibit reduced fitness in the presence of *Enterobacter cloacae* due to increased susceptibility to the type VI secretion system (T6SS). This finding is significant as it highlights the potential for interbacterial antagonism to shape the evolution of phage resistance. The authors further show that the O-antigen of lipopolysaccharide (LPS) plays a crucial role in protecting *S. enterica* from T6SS attack. This observation provides mechanistic insights into the fitness trade-offs associated with phage resistance.

The study's strength lies in its elegant experimental design and the comprehensive analysis of the interplay between phage resistance, T6SS susceptibility, and O-antigen structure. The authors employ a combination of co-culture experiments, genetic manipulations, and structural analyses to dissect the underlying mechanisms. The findings are robust and have implications for understanding the evolution of bacterial communities in the presence of phages and competing bacterial species.

This research will be of interest to a broad audience, including researchers in microbiology, synthetic biology, and microbial ecology. The findings have implications for understanding the evolution of phage resistance, and the dynamics of microbial communities. The study's insights

into the role of the O-antigen in T6SS susceptibility could also inform the design of novel antimicrobial strategies.

My expertise is microbial physiology

3. How much time do you estimate the authors will need to complete the suggested revisions:

Estimated time to Complete Revisions (Required)

(Decision Recommendation)

Between 1 and 3 months

4. Review Commons values the work of reviewers and encourages them to get credit for their work. Select 'Yes' below to register your reviewing activity at Web of Science Reviewer Recognition Service (formerly Publons); note that the content of your review will not be visible on Web of Science.

No

Review #3

1. Evidence, reproducibility and clarity:

Evidence, reproducibility and clarity (Required)

****Summary:****

Tsai et al. describe LPS biosynthesis mutants arising in selection for phage resistance that increase susceptibility to T6SS-mediated interbacterial antagonism. Phage-derived LPS degrading enzymes also contribute to T6SS susceptibility, which may be due to weakening of the physical barrier of LPS. The mechanisms of this fitness trade-off are elucidated with well-executed and presented experiments.

****Major comments:****

- No major critiques.

****Minor comments:****

- Others have described two T6SS in *Enterobacter cloacae* ATCC 13047 (PMID 33072020). Please clarify which of the two are inactivated by the *tssM* deletion in this study and either provide compelling evidence that both are inactive or change the text throughout to indicate T6SS-1 or T6SS-2 being inactivated.

- It seems the authors used EHEC EDL933, which has T6SS, in co-culture experiments (Figure 1C). Why do the authors think the *S. enterica* LPS mutants don't have a competitive disadvantage against EHEC? It seems to run counter to the conclusion that LPS is broadly protective against T6SS.

- It's not clear if the only Felix O1 and P22 phage-resistant transposon hits were in LPS-related genes, or if that pattern was observed in a more complete transposon sequencing dataset and selected for further study. A complete list of the sequence-identified hits, including the non-LPS related variants, would help clarify this and provide a useful resource to the research community.

- The fact that 8 of the 9 Felix O1 resistant variants all have transposon insertions in *waaO* should be stated in the results. The initial impression of showing R1-R9 is that 9 disrupted genes are being tested - in this case it's really only two. This is a minor critique because clean deletions by allelic exchange are shown for a more extensive set of genes anyway.

- The *S. enterica* serovar Typhimurium transposon mutagenesis library could benefit from clarification on details. The results section suggests use of a pre-existing "established" transposon library, but the methods and Figure 1 seem to indicate a new library was created based on prior methods. In either case, what is the genome coverage and redundancy of the library? If this is not known or saturation is not reached, the implications of potentially missing phage resistance genes with this approach should be discussed.

- There is some variation in phenotype among the strains with transposon insertion into the same gene, such as P22 resistant strain R7 which macroscopically agglutinates while the other *waaJ* insertions R5 and R1 don't. Is this due to polar effects on *waaO*, or could it be genetic alterations at other sites driven by stringent phage selection?

- Figure S1- The graphs with 12 growth curves are difficult to decipher, and the error bars would suggest maybe there are subtle growth differences among the mutants. Quantifying curve parameter(s) and applying a statistical test may clarify. The CFU counts in panel D seem to be not in log scale. Likewise in Figure S3 panel A, the authors say there are no significant growth defects, but the growth curves are modestly right-shifted for several mutants. This is a point of precision rather than a major critique, because the reversal of competitive growth phenotypes by donor T6SS inactivation indicate the potential minor growth defects aren't playing a major role in competition.

- Figure 3f - The authors say *fepE* is responsible for very long O-antigen chains, but it is not clear that the delta *fepE* LPS PAGE differs from wild type, which would fit with the lack of competitive disadvantage against *E. cloacae* in Figure 3g. The increased VL-modal O-antigen upon *fepE* overexpression in Figure 3h and increase protection in competition (figure 3i) are convincing. Is there another pathway(s) compensating for *fepE* deletion?

- Lines 199-200 - I believe the conclusion from *wzzB* deletion would be that L-modal O-antigen is necessary for protection against T6SS, and not necessarily sufficient.

- Do the environmentally isolated phages As2 and As4 encode TSP homologs?

2. Significance:

Significance (Required)

This manuscript provides a substantial advance in the field's understanding of how phages affect bacterial community interactions. To my knowledge, it is the first to bring together phage and T6SS defense with a strong mechanistic link. It's a conceptual advance in this regard that will stimulate more thought and experimentation on the roles of phage in bacterial communities like gut and

environmental microbiomes. The manuscript's strengths include rigorous overall design, clarity of the communication, and depth of mechanistic investigation, all the way down to atomic force microscopy measurements. There are some minor revisions suggested, but these are addressable with minimal/no additional experiments.

As someone with expertise in bacterial secretion systems and interbacterial interactions, I think this work will be of interest to microbiologists generally, and specifically in the fields of phage biology, bacterial secretion systems, and microbiome research. While the phage virology components are straightforward and well described, I think a review from someone with more expertise in this specific area would be beneficial.

3. How much time do you estimate the authors will need to complete the suggested revisions:

Estimated time to Complete Revisions (Required)

(Decision Recommendation)

Between 1 and 3 months

Yes

Full Revision

Manuscript number: RC-2024-02755

Corresponding author(s): See-Yeun, Ting

Point-by-point description of the revisions

Reviewer comments in **BLACK** and author responses in **BLUE**.

Reviewer #1 (Evidence, reproducibility, and clarity)

This manuscript by Tsai et al. shows that phage resistance mutations (LPS truncation) confer a cost during interbacterial competition. The authors show that various phage resistant mutants of *S. enterica* are inhibited by *E. cloacae* in a contact-dependent manner (on a solid surface but not in liquid). Further experiments showed that this inhibition of *S. enterica* was mediated by T6SS in *E. cloacae*. The authors then dissect which parts of the LPS are required for resistance against T6SS attacks and show that a similar resistance is conferred against T6SS of *B. thailandensis* and *C. rodentium*. Moreover, the authors show that enzymatic degradation of LPS by a phage enzyme can also increase sensitivity to T6SS (including when such enzymes are on phage particles). Finally, the authors suggest that the change in the thickness of the LPS surface layer could be the reason for changes in T6SS susceptibility. Overall, the manuscript is very well-written. The experiments and controls are explained in sufficient detail and in a logical order. The figures are clear and easy to navigate. The findings are very interesting and important for the T6SS field but also for general understanding how different evolutionary pressures combine and influence each other. I believe that this manuscript will initiate further research in this direction.

We thank the reviewer for their positive remarks on our manuscript and the valuable suggestions for its improvement.

Major comments

The only major point that I would like to raise is that I am not generally convinced that the 2 nm difference in the thickness of LPS is the main reason for the observed differences in T6SS-mediated killing of *S. enterica*. Based on what we know about T6SS mode of action, we expect that it is potentially pushing effectors by up to several hundreds of nanometers. Therefore, the change in the LPS thickness by a few nanometers (as measured by AFM) seems insufficient to provide enough spacing between the attacker and the prey to significantly decrease T6SS effector delivery. While it is clear that understanding the exact reason for the LPS mediated resistance is beyond the scope of this manuscript, I would suggest that the authors consider the fact that T6SS is known to deliver proteins even to the cytoplasm of target gram-negative cells and discuss the mode of action of the machine in the context of their finding. If the T6SS was drawn to scale in the model figure, it would become apparent that 2 nm change in the distance between two cells has probably no major impact on killing by T6SS and the actual reason for the observed phenotype is likely more complicated than what is proposed.

We appreciate the reviewer's comments and acknowledge that our manuscript leaves open questions regarding the exact mechanisms underlying LPS-mediated resistance. We have now moderated the Discussion in our revised manuscript to reflect the complexity of this phenomenon (**Lines 410-423**). Although we agree that the nanometer difference in LPS thickness may not fully explain the observed protective phenotype, we believe it remains a plausible contributing factor that is worth considering.

To fully understand how LPS influences T6SS effector delivery, future studies will need to address key mechanistic questions regarding the T6SS injection process. For example, 1) how deeply does the T6SS apparatus penetrate the target Gram-negative cells during injection; 2) what is the magnitude of the injection force generated by the T6SS; and 3) does the structural integrity of the T6SS apparatus remain intact throughout and after contraction? While it is well documented that some T6SS effectors act in the cytosol of target cells, there is evidence to suggest that cytosolic effectors are initially delivered into the periplasm and subsequently translocated into the cytosol for intoxication^{1,2}. Furthermore, although contraction of the T6SS apparatus occurs within milliseconds^{3,4}, this rapid action does not preclude the possibility that the injection force could be influenced by the thickness of the LPS layer. In addition, the stability of T6SS structural or delivered proteins—such as PAAR, VgrG, and Hcp—within the delivery complex might be compromised upon encountering physical barriers such as the LPS layer and the outer membrane of target cells. These potential interactions could affect the efficiency of effector delivery, leading to reduced competitiveness during interbacterial antagonism, as shown in our study.

We appreciate the reviewer's suggestions and acknowledge that the precise reasons for LPS-mediated resistance likely involve a combination of factors beyond those proposed here. We are actively pursuing these questions as part of an ongoing, long-term effort to better elucidate the mechanisms of T6SS action.

Minor comments

Specify which T6SS of *B. thailandensis* was tested.

We now cite studies by Schwarz, S., et al., 2010⁵ and LeRoux, M., et al., 2015⁶, from which we used the *tssM* (BTH_I2954) gene deletion strain abrogating the T6SS-1 of the *B. thailandensis* E264 (**Line 234, Supplementary Table 1**).

Use a different naming of the two strains used in competition assays than "donor" and "recipient".

Thank you for this suggestion. In the revised manuscript, we have replaced the terms "donor" and "recipient" with "attacker" and "prey" for clarity. This change has been applied to the text (**Lines 441, and 649-667**) and to **revised Figures 2c-h, Figures 3b, d, g, i, j, Figures 4f, g, Figures 5b, e, g, h, Supplementary Figures 3d-f, and Supplementary Figures 4b-d**.

Indicate in the material and methods ODs of bacterial mixtures used in the "Bacterial competition assays".

We apologize for this oversight. The ODs of bacterial mixtures used in the "Bacterial competition assays" have now been specified in the revised Methods section (**Line 651**).

Reviewer #1 (Significance)

This manuscript is interesting for researchers who study T6SS, phage predation and other evolutionary pressures shaping bacterial interactions. The work provides new and interesting insights. My expertise in LPS biology is limited.

We sincerely appreciate the reviewer's interest in and support of our study.

Reviewer #2 (Evidence, reproducibility, and clarity)

This work investigates the fitness trade-offs in *Salmonella enterica* resistant to phages. The authors performed co-culture experiments with *S. enterica*, *E. coli*, and *E. cloacae* and found that phage-resistant *S. enterica* strains displayed reduced fitness in the presence of *E. cloacae*. Further experiments demonstrated that phage-resistant *S. enterica* strains were more susceptible to the type VI secretion system (T6SS) of *E. cloacae*. The authors then examined the role of the O-antigen of lipopolysaccharide (LPS) in T6SS-mediated interbacterial antagonism. By constructing *S. enterica* mutants with varying O-antigen chain lengths, the authors demonstrated that the O-antigen protects *S. enterica* from T6SS attack. They then demonstrated that the O-antigen-deficient *S. enterica*, *E. coli*, and *C. rodentium* strains were more susceptible to T6SS attack by *E. cloacae*. Finally, the authors showed that phage tail spike proteins (TSPs) with endoglycosidase activity could cleave the bacterial O-antigen, thereby increasing susceptibility to T6SS attack.

The study is well-designed and the experiments are well-executed. The findings are significant and have implications for the understanding of microbial community dynamics.

We thank the reviewer for their positive comments regarding our original submission.

Major comments

While the study elegantly demonstrates the link between phage resistance, LPS structure, and T6SS susceptibility, we must remember that these LPS-defective strains are likely at a significant disadvantage in real-world environments without the influence of competing bacteria. Whether it's the gut or external environments, *Salmonella* needs its LPS for protection against a myriad of host and environmental factors. It seems a bit redundant for T6SS mediated antagonism to select for LPS structures when those structures are essential for bacterial survival outside of this very specific context. It would benefit some discussion about the likelihood of these phage-resistant, LPS-defective strains actually persisting and competing effectively in a more natural setting.

We thank the reviewer for their insightful comments and appreciate the opportunity to clarify this point. We agree that LPS-defective bacterial strains face significant disadvantages in natural environments, where they must contend with various host and environmental stresses. Consequently, we did not intend to suggest that T6SS-mediated antagonism is the primary driving force in selecting specific LPS structures. Rather, our study highlights an additional role for LPS during interbacterial interactions, complementing its well-established functions. This notion aligns with the hypotheses proposed in prior studies⁷⁻⁹.

The reviewer's comments raise an intriguing question about the essentiality of LPS in Gram-negative bacteria under natural conditions. During our revision process, we identified several examples in the literature demonstrating that LPS may not always be indispensable. For instance, LPS-depleted *Neisseria meningitidis* strains with an early block in lipid A biosynthesis have been shown to remain viable^{10,11}. These strains may possess adaptive advantages under specific circumstances¹². Similarly, some pathogenic bacteria produce truncated LPS structures lacking O-antigen or introduce modified LPS to evade host immune responses¹³. Additionally, evolutionary pressures, such as phage predation, often drive mutations in O-antigen biosynthesis pathways, resulting in alterations to or an absence of O-antigen¹⁴. Furthermore, recent studies have also indicated

that trade-offs between abiotic and biotic stresses can influence LPS integrity. For instance, LPS-deficient strains may exhibit selective advantages in extreme environments^{15,16}. These findings underscore the context-dependent nature of LPS functionality and its potential dispensability in certain ecological niches.

We sincerely appreciate the reviewer's thought-provoking comments. Our current study aims to provide evidence for the role of interbacterial antagonism as an additional factor influencing LPS integrity. However, we did not mean to overstate the contribution of this mechanism. Instead, we only seek to contribute to a broader understanding of the multifaceted functions of LPS in bacterial survival and adaptation. We have modified the Discussion in our revised manuscript to clarify this idea (**Lines 453-466**).

Minor comments

Figure 5 could be more effective if panels b and c are together

We appreciate this suggestion. We have revised the manuscript accordingly, so panels *b* and *c* have been combined in **revised Figure 5**, and the respective **figure legends** have been modified for improved clarity (**Lines 810-823**).

#69 Authors should define mucoid

The term "mucoid" has now been defined in the revised manuscript (**Lines 69-70**).

#155 Authors should explain that this result is expected since T6SS acts on solid surface while CDI works in liquid cultures

Thank you for this comment. Prior studies have demonstrated that while CDI-mediated antibacterial activity is less efficient in liquid environments, it can still occur on both solid surfaces and in liquid cultures, provided the competitors possess the necessary CdiA binding unit, such as BamA^{17,18}. This understanding supports our initial hypothesis that T6SS and/or CDI contribute to the observed protective phenotype in *S. enterica* phage-resistant variants (**Figure 2**).

#clarify what it is meant by unicellular cultures. Should it be monocultures?

We apologize for this error and have now replaced "unicellular cultures" with "monocultures" in the revised manuscript (**Lines 137, 180, and 258**).

#618 add to the text how much dead phage was added per bacterial cell

Apologies for this oversight. The multiplicity of infection (MOI) describing the amount of inactivated phages used to treat bacterial cells has now been included in the revised Methods section (**Line 661**).

#364 references needed for "consistent with predictions for intact LPS structures "

We thank the reviewer for pointing out this omission. The relevant reference has now been added to the revised manuscript¹⁹ (**Line 368**).

Reviewer #2 (Significance)

This study offers a new perspective on the interplay between phage resistance and bacterial fitness in the context of microbial communities. While the concept of fitness trade-offs associated with antibiotic resistance is well-established, the authors extend this paradigm to phage resistance. They demonstrate that phage-resistant *Salmonella enterica* strains exhibit reduced fitness in the presence of *Enterobacter cloacae* due to increased susceptibility to the type VI secretion system (T6SS). This finding is significant as it highlights the potential for interbacterial antagonism to shape the evolution of phage resistance. The authors further show that the O-antigen of lipopolysaccharide (LPS) plays a crucial role in protecting *S. enterica* from T6SS attack. This observation provides mechanistic insights into the fitness trade-offs associated with phage resistance.

The study's strength lies in its elegant experimental design and the comprehensive analysis of the interplay between phage resistance, T6SS susceptibility, and O-antigen structure. The authors employ a combination of co-culture experiments, genetic manipulations, and structural analyses to dissect the underlying mechanisms. The findings are robust and have implications for understanding the evolution of bacterial communities in the presence of phages and competing bacterial species.

This research will be of interest to a broad audience, including researchers in microbiology, synthetic biology, and microbial ecology. The findings have implications for understanding the evolution of phage resistance, and the dynamics of microbial communities. The study's insights into the role of the O-antigen in T6SS susceptibility could also inform the design of novel antimicrobial strategies.

My expertise is microbial physiology

We thank the reviewer for their positive remarks and careful reading of our manuscript.

Reviewer #3 (Evidence, reproducibility, and clarity)

Tsai et al. describe LPS biosynthesis mutants arising in selection for phage resistance that increase susceptibility to T6SS-mediated interbacterial antagonism. Phage-derived LPS degrading enzymes also contribute to T6SS susceptibility, which may be due to weakening of the physical barrier of LPS. The mechanisms of this fitness trade-off are elucidated with well-executed and presented experiments.

We are grateful to the reviewer for their kind words and critical reading of the manuscript.

Major comments

- No major critiques.

Minor comments

- Others have described two T6SS in *Enterobacter cloacae* ATCC 13047 (PMID 33072020). Please clarify which of the two are inactivated by the *tssM* deletion in this study and either provide compelling evidence that both are inactive or change the text throughout to indicate T6SS-1 or T6SS-2 being inactivated.

We thank the reviewer for this comment. In our study, we refer to the work by Whitney, J., et al., 2014²⁰, from which we used the *tssM* (ECL_01536) gene deletion strain in which T6SS-1 of the *E. cloacae* ATCC 13047 is abrogated. Consistent with this detail, we have now clarified in the revised manuscript (**Line 155, Supplementary Table 1**) that T6SS-1 is inactivated. Moreover, the reference suggested by the reviewer provides additional evidence supporting that T6SS-1, but not T6SS-2, is involved in bacterial competition²¹, which we also now specify in the revised manuscript.

- It seems the authors used EHEC EDL933, which has T6SS, in co-culture experiments (Figure 1C). Why do the authors think the *S. enterica* LPS mutants don't have a competitive disadvantage against EHEC? It seems to run counter to the conclusion that LPS is broadly protective against T6SS.

We thank the reviewer for raising this point. While it is true that EHEC O157:H7 strain EDL933 possesses a T6SS gene cluster in its genome, a prior study has shown that the T6SS in this strain appears to be inactivated under laboratory conditions, likely due to repression by the global regulator H-NS²². Consistent with these findings, our data indicate that the *S. enterica* LPS mutants did not exhibit a competitive disadvantage against EHEC EDL933. These results support the conclusion that, under the conditions tested, the truncated LPS in *S. enterica* does not affect its fitness against EHEC (**Figure 1c**), likely due to the inactivity of the EHEC T6SS²².

- It's not clear if the only Felix O1 and P22 phage-resistant transposon hits were in LPS-related genes, or if that pattern was observed in a more complete transposon sequencing dataset and selected for further study. A complete list of the sequence-identified hits, including the non-LPS related variants, would help clarify this and provide a useful resource to the research community.

We thank the reviewer for the opportunity to clarify this point. For each phage, we initially isolated nine phage-resistant transposon variants, which were subsequently used for co-culture assays and transposon insertion site identification, as described in the original manuscript (**Figure 1a** and **Supplementary Figure 2a**).

We agree with the reviewer that a broader screening approach could reveal non-LPS-related variants and provide a more comprehensive resource for the research community. To address this point, during the manuscript revision period, we followed the same procedure and isolated an additional nine phage-resistant variants for each phage (**Supplementary Table 1**). Interestingly, from this expanded isolation dataset, the transposon insertions were again found exclusively in LPS-related genes (**Author Response Figure 1**).

We have now included this new dataset in the revised manuscript and believe it strengthens the robustness of our findings. This expanded data has been made available below for further reference:

Author Response Figure 1.

Characterization of additional transposon insertion mutations in phage-resistant *S. enterica* isolates. Mutated genes in the P22-resistant and Felix O1-resistant *S. enterica* isolates. Scale bar = 500 bp.

- The fact that 8 of the 9 Felix O1 resistant variants all have transposon insertions in *waaO* should be stated in the results. The initial impression of showing R1-R9 is that 9 disrupted genes are being tested - in this case it's really only two. This is a minor critique because clean deletions by allelic exchange are shown for a more extensive set of genes anyway.

We thank the reviewer for this comment. As suggested, we have revised the Results section (**Lines 126-131**) to explicitly state that Felix O1-resistant variants harbor transposon insertions in only two genes (*waaO* and *dagR*), which were initially tested in the competition assay (**Figure 2**).

- The *S. enterica* serovar Typhimurium transposon mutagenesis library could benefit from clarification on details. The results section suggests use of a pre-existing "established" transposon library, but the methods and Figure 1 seem to indicate a new library was created based on prior methods. In either case, what is the genome coverage and redundancy of the library? If this is not known or saturation is not reached, the implications of potentially missing phage resistance genes with this approach should be discussed.

We thank the reviewer for the opportunity to clarify this point. For our study, we created a transposon library following previously established methods²³. The library comprises approximately 12,000 variants, as noted in

Figure 1a. While doing so provided substantial genome coverage, it did not achieve full saturation. We have now revised the Results section (**Lines 93-94, and 115-117**) to better describe the potential limitations of this approach, including by stating the possibility that some phage-resistance genes may have been missed during the screening.

- There is some variation in phenotype among the strains with transposon insertion into the same gene, such as P22 resistant strain R7 which macroscopically agglutinates while the other *waaJ* insertions R5 and R1 don't. Is this due to polar effects on *waaO*, or could it be genetic alterations at other sites driven by stringent phage selection?

We thank the reviewer for this comment. We also suspect that the variation in the macroscopically agglutinative phenotypes among P22-resistant strains, such as strain R7 compared to R5 and R1, may be caused by polar effects on *waaO*. Additionally, the possibility of genetic alterations at other loci driven by stringent phage selection cannot be excluded.

To address this potential variability and ensure consistency, we used clean deletions of each LPS biogenesis gene in all subsequent experiments. This approach eliminates the confounding effects of polar mutations or secondary genetic alterations, thereby providing more robust and interpretable data.

- Figure S1- The graphs with 12 growth curves are difficult to decipher, and the error bars would suggest maybe there are subtle growth differences among the mutants. Quantifying curve parameter(s) and applying a statistical test may clarify. The CFU counts in panel D seem to be not in log scale. Likewise in Figure S3 panel A, the authors say there are no significant growth defects, but the growth curves are modestly right-shifted for several mutants. This is a point of precision rather than a major critique, because the reversal of competitive growth phenotypes by donor T6SS inactivation indicate the potential minor growth defects aren't playing a major role in competition.

We thank the reviewer for these suggestions and corrections. We have now revised the manuscript accordingly, including in **Supplementary Figures 1 and 3**. Quantitative analysis of growth curve parameters and statistical tests have been included below to clarify the observed differences (**Author Response Figure 2**).

The slight right-shift of the growth curves for some mutants, as noted in **Supplementary Figure 3**, may be attributable to cell aggregation, as shown in **Supplementary Figures 2e, f**. The growth rate measurements were conducted in a 96-well plate with steady shaking at 200 rpm using a plate reader, which does not fully account for the aggregated cell phenotype. Despite these subtle growth differences, we agree with the reviewer that they do not appear to play a major role in the competitive growth phenotypes, as evidenced by the reversal of phenotypes upon donor T6SS inactivation (**Supplementary Figure 3**).

Author Response Figure 2.

Statistical analysis of bacterial growth curves of *S. enterica* strains obtained by fitted to Gompertz equation.
 (a, b) Statistical growth rate (a) or lag time (b) of growth curves of the indicated phage-resistant *S. enterica* strains (Supplementary Figure 1c). Wild-type, $\Delta waaL$, and $\Delta waaG$ strains are shown as controls. (c, d) Statistical growth rate (c) or lag time (d) of growth curves of the indicated *S. enterica* LPS mutants (Supplementary Figure 3a-c). (e, f) Statistical growth rate (e) or lag time (f) of growth curves of the indicated *S. enterica* incubated with 10 nM of purified TSP (wild-type or D392N mutant) (Supplementary Figure 5b).

• Figure 3f - The authors say *fepE* is responsible for very long O-antigen chains, but it is not clear that the delta *fepE* LPS PAGE differs from wild type, which would fit with the lack of competitive disadvantage against *E. cloacae* in Figure 3g. The increased VL-modal O-antigen upon *fepE* overexpression in Figure 3h and increase protection in competition (figure 3i) are convincing. Is there another pathway(s) compensating for *fepE* deletion?

We thank the reviewer for this thoughtful comment. We have repeated the experiment independently at least three times and consistently observed a reduction in the VL-modal O-antigen in the $\Delta fepE$ strain. To provide additional clarity, we have included supplementary LPS profiles and quantifications below (Author Response Figure 3). We currently do not have evidence from the literature or our experiments to identify an alternative pathway compensating for the deletion of *fepE*. Nonetheless, we acknowledge this as a possibility and appreciate the reviewer's insight into this topic.

Author Response Figure 3.

Quantification of the VL- and L-modal O-antigens in the indicated *S. enterica* variants.

(a) 12% LPS PAGE profiles showing LPS levels produced by the indicated *S. enterica* strains. (b) The associated histogram of the densitometry scans for each lane of the LPS PAGE, illustrating the regions for densitometry analysis. (c) Densitometry analysis of the above LPS PAGEs. The ratio of VL-modal O-Ag or L-modal O-Ag density to that of lipid A were compared between *S. enterica* wild-type strain with various mutants.

Full Revision

- Lines 199-200 - I believe the conclusion from wzzB deletion would be that L-modal O-antigen is necessary for protection against T6SS, and not necessarily sufficient.

We thank the reviewer for pointing out this important distinction. The respective sentence has now been revised in the manuscript (**Line 204**).

- Do the environmentally isolated phages As2 and As4 encode TSP homologs?

We thank the reviewer for this question. We did not identify TSP homologs in the genome of As2 and As4 phages. The genome sequences of As1 to As4 have been uploaded to NCBI's BioProject resource under accession number PRJNA1199570 (**Lines 535-544, 741-743**).

Reviewer #3 (Significance)

This manuscript provides a substantial advance in the field's understanding of how phages affect bacterial community interactions. To my knowledge, it is the first to bring together phage and T6SS defense with a strong mechanistic link. It's a conceptual advance in this regard that will stimulate more thought and experimentation on the roles of phage in bacterial communities like gut and environmental microbiomes. The manuscript's strengths include rigorous overall design, clarity of the communication, and depth of mechanistic investigation, all the way down to atomic force microscopy measurements. There are some minor revisions suggested, but these are addressable with minimal/no additional experiments.

As someone with expertise in bacterial secretion systems and interbacterial interactions, I think this work will be of interest to microbiologists generally, and specifically in the fields of phage biology, bacterial secretion systems, and microbiome research. While the phage virology components are straightforward and well described, I think a review from someone with more expertise in this specific area would be beneficial.

We thank the reviewer for their careful reading of our manuscript and for the suggestions to improve it.

References

1. Whitney, J.C., Quentin, D., Sawai, S., LeRoux, M., Harding, B.N., Ledvina, H.E., Tran, B.Q., Robinson, H., Goo, Y.A., Goodlett, D.R., et al. (2015). An interbacterial NAD(P)(+) glycohydrolase toxin requires elongation factor Tu for delivery to target cells. *Cell* *163*, 607-619. 10.1016/j.cell.2015.09.027.
2. Ali, J., Yu, M., Sung, L.K., Cheung, Y.W., and Lai, E.M. (2023). A glycine zipper motif is required for the translocation of a T6SS toxic effector into target cells. *EMBO Rep* *24*, e56849. 10.15252/embr.202356849.
3. LeRoux, M., De Leon, J.A., Kuwada, N.J., Russell, A.B., Pinto-Santini, D., Hood, R.D., Agnello, D.M., Robertson, S.M., Wiggins, P.A., and Mougous, J.D. (2012). Quantitative single-cell characterization of bacterial interactions reveals type VI secretion is a double-edged sword. *Proc Natl Acad Sci U S A* *109*, 19804-19809. 10.1073/pnas.1213963109.
4. Basler, M., Pilhofer, M., Henderson, G.P., Jensen, G.J., and Mekalanos, J.J. (2012). Type VI secretion requires a dynamic contractile phage tail-like structure. *Nature* *483*, 182-186. 10.1038/nature10846.
5. Schwarz, S., West, T.E., Boyer, F., Chiang, W.C., Carl, M.A., Hood, R.D., Rohmer, L., Tolker-Nielsen, T., Skerrett, S.J., and Mougous, J.D. (2010). Burkholderia type VI secretion systems have distinct roles in eukaryotic and bacterial cell interactions. *PLoS Pathog* *6*, e1001068. 10.1371/journal.ppat.1001068.
6. LeRoux, M., Kirkpatrick, R.L., Montauti, E.I., Tran, B.Q., Peterson, S.B., Harding, B.N., Whitney, J.C., Russell, A.B., Traxler, B., Goo, Y.A., et al. (2015). Kin cell lysis is a danger signal that activates antibacterial pathways of *Pseudomonas aeruginosa*. *Elife* *4*. 10.7554/eLife.05701.
7. Hersch, S.J., Manera, K., and Dong, T.G. (2020). Defending against the Type Six Secretion System: beyond Immunity Genes. *Cell Rep* *33*, 108259. 10.1016/j.celrep.2020.108259.
8. Unterweger, D., Kitaoka, M., Miyata, S.T., Bachmann, V., Brooks, T.M., Moloney, J., Sosa, O., Silva, D., Duran-Gonzalez, J., Provenzano, D., and Pukatzki, S. (2012). Constitutive type VI secretion system expression gives *Vibrio cholerae* intra- and interspecific competitive advantages. *PLoS One* *7*, e48320. 10.1371/journal.pone.0048320.
9. Toska, J., Ho, B.T., and Mekalanos, J.J. (2018). Exopolysaccharide protects *Vibrio cholerae* from exogenous attacks by the type 6 secretion system. *Proc Natl Acad Sci U S A* *115*, 7997-8002. 10.1073/pnas.1808469115.
10. Steeghs, L., den Hartog, R., den Boer, A., Zomer, B., Roholl, P., and van der Ley, P. (1998). Meningitis bacterium is viable without endotoxin. *Nature* *392*, 449-450. 10.1038/33046.
11. Steeghs, L., de Cock, H., Evers, E., Zomer, B., Tommassen, J., and van der Ley, P. (2001). Outer membrane composition of a lipopolysaccharide-deficient *Neisseria meningitidis* mutant. *EMBO J* *20*, 6937-6945. 10.1093/emboj/20.24.6937.
12. Fransen, F., Heckenberg, S.G., Hamstra, H.J., Feller, M., Boog, C.J., van Putten, J.P., van de Beek, D., van der Ende, A., and van der Ley, P. (2009). Naturally occurring lipid A mutants in *neisseria meningitidis* from patients with invasive meningococcal disease are associated with reduced coagulopathy. *PLoS Pathog* *5*, e1000396. 10.1371/journal.ppat.1000396.
13. Maldonado, R.F., Sa-Correia, I., and Valvano, M.A. (2016). Lipopolysaccharide modification in Gram-negative bacteria during chronic infection. *FEMS Microbiol Rev* *40*, 480-493. 10.1093/femsre/fuw007.
14. Yu, J., Zhang, H., Ju, Z., Huang, J., Lin, C., Wu, J., Wu, Y., Sun, S., Wang, H., Hao, G., and Zhang, A. (2024). Increased mutations in lipopolysaccharide biosynthetic genes cause time-dependent development of phage resistance in *Salmonella*. *Antimicrob Agents Chemother* *68*, e0059423. 10.1128/aac.00594-23.
15. Burmeister, A.R., Fortier, A., Roush, C., Lessing, A.J., Bender, R.G., Barahman, R., Grant, R., Chan, B.K., and Turner, P.E. (2020). Pleiotropy complicates a trade-off between phage resistance and antibiotic resistance. *Proc Natl Acad Sci U S A* *117*, 11207-11216. 10.1073/pnas.1919888117.

16. Carretero-Ledesma, M., Garcia-Quintanilla, M., Martin-Pena, R., Pulido, M.R., Pachon, J., and McConnell, M.J. (2018). Phenotypic changes associated with Colistin resistance due to Lipopolysaccharide loss in *Acinetobacter baumannii*. *Virulence* 9, 930-942. 10.1080/21505594.2018.1460187.
17. Aoki, S.K., Pamma, R., Hernday, A.D., Bickham, J.E., Braaten, B.A., and Low, D.A. (2005). Contact-dependent inhibition of growth in *Escherichia coli*. *Science* 309, 1245-1248. 10.1126/science.1115109.
18. Aoki, S.K., Malinverni, J.C., Jacoby, K., Thomas, B., Pamma, R., Trinh, B.N., Remers, S., Webb, J., Braaten, B.A., Silhavy, T.J., and Low, D.A. (2008). Contact-dependent growth inhibition requires the essential outer membrane protein BamA (YaeT) as the receptor and the inner membrane transport protein AcrB. *Mol Microbiol* 70, 323-340. 10.1111/j.1365-2958.2008.06404.x.
19. Gao, Y., Widmalm, G., and Im, W. (2023). Modeling and Simulation of Bacterial Outer Membranes with Lipopolysaccharides and Capsular Polysaccharides. *J Chem Inf Model* 63, 1592-1601. 10.1021/acs.jcim.3c00072.
20. Whitney, J.C., Beck, C.M., Goo, Y.A., Russell, A.B., Harding, B.N., De Leon, J.A., Cunningham, D.A., Tran, B.Q., Low, D.A., Goodlett, D.R., et al. (2014). Genetically distinct pathways guide effector export through the type VI secretion system. *Mol Microbiol* 92, 529-542. 10.1111/mmi.12571.
21. Soria-Bustos, J., Ares, M.A., Gomez-Aldapa, C.A., Gonzalez, Y.M.J.A., Giron, J.A., and De la Cruz, M.A. (2020). Two Type VI Secretion Systems of *Enterobacter cloacae* Are Required for Bacterial Competition, Cell Adherence, and Intestinal Colonization. *Front Microbiol* 11, 560488. 10.3389/fmicb.2020.560488.
22. Wan, B., Zhang, Q., Ni, J., Li, S., Wen, D., Li, J., Xiao, H., He, P., Ou, H.Y., Tao, J., et al. (2017). Type VI secretion system contributes to Enterohemorrhagic *Escherichia coli* virulence by secreting catalase against host reactive oxygen species (ROS). *PLoS Pathog* 13, e1006246. 10.1371/journal.ppat.1006246.
23. Mandal, R.K., Jiang, T., and Kwon, Y.M. (2021). Genetic Determinants in *Salmonella enterica* Serotype Typhimurium Required for Overcoming In Vitro Stressors in the Mimicking Host Environment. *Microbiol Spectr* 9, e0015521. 10.1128/Spectrum.00155-21.

Dr. See-Yeun Ting
Institute of Molecular Biology, Academia Sinica, Taipei, Taiwan
Taiwan

14th Feb 2025

Re: EMBOJ-2024-120021-T
Surface-mediated Bacteriophage Defense Incurs Fitness Tradeoffs for Interbacterial Antagonism

Dear Dr. Ting,

Thank you for submitting your revised Review Commons preprint to The EMBO Journal. Given the interest of the study and overall fit within our journal's scope, we decided to treat it like a regular revision, and returned it to two of the original referees. As you will see from their comments copied below, both of them were fully satisfied with the revisions, and we shall therefore be happy to publish this work in The EMBO Journal, following correction of the following editorial and format issues:

- Please download (see link below) our author checklist, and upload it in completed form with the final manuscript.
- Please upload the manuscript text (including figure legends) as an editable text file, and all figures without legends as individual image files with sufficient resolution/quality for production.
- Please adjust the order of the manuscript sections: Title page with complete author information, Abstract, Keywords, Introduction, Results, Discussion, Methods, Data Availability, Acknowledgements, Disclosure and Competing Interests Statement, References, Main Figure Legends, Tables, Expanded Figure Legends.
- Please rename the Conflict of Interest section into "Disclosure and Competing Interests Statement", in accordance with our updated Guide to Authors (<https://www.embopress.org/competing-interests>)
- As we are switching from a free-text author contribution statement towards a more formal statement based on Contributor Role Taxonomy (CRediT) terms, please remove the present Author Contribution section and instead specify each author's contribution(s) directly in the Author Information page of our submission system during upload of the final manuscript. See <https://casrai.org/credit/> for more information.
- Please adjust the format of the reference list and of the in-text citations according to EMBO Journal format (alphabetical order, author name et al + year...)
- Please note that Materials and Methods need to be described in the main text using our 'Structured Methods' format (for detail, see <https://www.embopress.org/page/journal/14693178/authorguide#structuredmethods>). The in-text "Methods" section should contain method and protocol descriptions (ideally using a step-by-step protocol format to facilitate adoption of the methodologies across labs), while all key reagents, experimental models, software and relevant equipment - including their sources and relevant identifiers - should be listed in a separately uploaded Reagents and Tools Table, a template for which can be downloaded from the above-linked section of our Author Guidelines.
- Please make sure to call-out each sub-panel of each figure at least once (and in order). E.g., for Figure 7, there is currently only a reference to the whole figure (which might need to be changed e.g. to "Fig 7A-F").
- In the Data Availability section, please include a direct hyperlink to the database in which the deposited data can be accessed.
- Please rename the "supplementary figures" into Expanded View Figures (call-out: "Figure EV1/2/..."), both in the legends and when referencing them in the text. See www.embopress.org/page/journal/14602075/authorguide#expandedview for further information.
- Please rename the "supplementary table" as Expanded View Table (reference: "Table EV1"), and upload it as a separate Word or Excel file.
- Please provide suggestions for a short 'blurb' text prefacing and summing up the study in two sentences (max. 250 characters), followed by 3-5 one-sentence 'bullet points' with brief factual statements about key results of the paper; they will form the basis of an editor-written 'Synopsis' accompanying the online version of the article (see new articles on our journal website for some recent examples). Please also provide a simple synopsis image, which can be used as a "visual title" for the synopsis section of your paper (maybe a simplified/compacted version of Figure 7?). The image should be in PNG or JPG format with the modest dimensions of 550 x 300-600 pixels (width x height).

- During routine pre-acceptance checks, our data editors have raised the following queries regarding figures, data, and legends, which I would ask you to address (ideally using the Track Changes option):

1. Please define the annotated p values ****/***/**/* as well as provide the exact p-values for the same in the legend of figure 6A, B, K as appropriate.
2. Please note that the exact p values are not provided in the legends of figures 1C, 2C, D, G, H; 3B, D, G, I, J; 4F, 5B, E, G.
3. Please indicate the statistical test used for data analysis in the legends of figures 1C, 2C, D, G, H; 3B, D, G, I, J; 4F, G; 5B, E, G.
4. Please note that the box plots need to be defined in terms of minima, maxima, centre, bounds of box and whiskers, and percentile in the legend of figure 6K
5. Please note that information related to n is missing in the legends of figures 6A, B, K
6. Please note that the error bars are not defined in the legends of figures 6A, B.

- Finally, please note the attached request for Source Data preparation and uploading from our SourceData Scientific Coordinator. The attached table contains a list of the figure panels for which we require source data, and this table has to be diligently completed and provided at the time of resubmission. Please note that we do not expect you to provide us with "raw data" (e.g., qPCR Ct values), but with minimally processed data underlying the qualitative and quantitative data summarized in and used to generate the figures (e.g., mRNA increased expression folds), including replicates.

I am therefore returning the manuscript to you now for a final round of minor revision, to allow you to make all these adjustments and upload all modified files. Once we will have received them, we should hopefully be able to swiftly proceed with acceptance and publication of the manuscript. Please do not hesitate to get back to us in case you should have any questions in this regard.

Yours sincerely,

Hartmut Vodermaier

Revision to The EMBO Journal should be submitted online within 90 days, unless an extension has been requested and approved by the editor; please click on the link below to submit the revision online before 15th May 2025:

Link Not Available

Referee #2:

The authors provided a thoughtful and comprehensive response to the reviewer's comments. My only major concern was the general disadvantage that may arise from the loss of LPS. The authors made their point with a thorough argument supported by the literature. Also, the authors made modifications to the manuscript to avoid misinterpretations on this topic. All my minor concerns were also addressed. Overall, this is a very exciting manuscript that provides a fresh perspective on a relevant topic.

Referee #3:

All significant concerns from the original review have been appropriately addressed. I recommend publication.

Rev_Com_number: RC-2024-02755
New_manu_number: EMBOJ-2024-120021-T
Corr_author: Ting
Title: Surface-mediated Bacteriophage Defense Incurs Fitness Tradeoffs for Interbacterial Antagonism

Referee #2:

The authors provided a thoughtful and comprehensive response to the reviewer's comments. My only major concern was the general disadvantage that may arise from the loss of LPS. The authors made their point with a thorough argument supported by the literature. Also, the authors made modifications to the manuscript to avoid misinterpretations on this topic. All my minor concerns were also addressed. Overall, this is a very exciting manuscript that provides a fresh perspective on a relevant topic.

We are grateful to the reviewer for their kind words on our manuscript.

Referee #3:

All significant concerns from the original review have been appropriately addressed. I recommend publication.

We thank the reviewer for their positive remarks on our manuscript.

Dr. See-Yeun Ting
Institute of Molecular Biology, Academia Sinica, Taipei, Taiwan
Taiwan

27th Feb 2025

Re: EMBOJ-2024-120021R
Surface-mediated Bacteriophage Defense Incurs Fitness Tradeoffs for Interbacterial Antagonism

Dear Dr. Ting,

Thank you for submitting your final revised manuscript for our consideration. I am pleased to inform you that we have now accepted it for publication in The EMBO Journal.

Yours sincerely,

Hartmut Vodermaier
